# Compounding inequalities: Adolescent psychosocial wellbeing and resilience among refugee and host communities in Jordan during the COVID-19 pandemic

Nicola Jones[1‡]*, Sarah Baird[2‡], Bassam Abu Hamad[3], Zulfiqar A. Bhutta[4,5], Erin Oakley[2], Manisha Shah[6], Jude Sajdi[7], Kathryn M. Yount[8]

1 Gender and Adolescence: Global Evidence (GAGE), ODI, London, United Kingdom, 2 Department of Global Health, George Washington University, Washington, DC., United States of America, 3 Department of Public Health, Al Quds University, Gaza, State of Palestine, 4 Department of Nutritional Sciences, Hospital for Sick Children, Toronto, Canada, 5 Aga Khan University, Karachi, Pakistan, 6 Luskin School of Public Affairs, University of California, Los Angeles, CA, United States of America, 7 Information and Research Center, King Hussein Foundation, Amman, Jordan, 8 Department of Global Health, Emory University, Atlanta, GA, United States of America

‡ NJ and SB are joint first authors on this work.
* n.jones@odi.org.uk

**Data Availability Statement:** The data underpinning the paper is publicly available from

## Abstract

### Purpose

The COVID-19 pandemic and associated risk-mitigation strategies have altered the social contexts in which adolescents in low- and middle-income countries live. Little is known, however, about the impacts of the pandemic on displaced populations, and how those impacts differ by gender and life stage. We investigate the extent to which the pandemic has compounded pre-existing social inequalities among adolescents in Jordan, and the role support structures play in promoting resilience.

### Methods

Our analysis leverages longitudinal quantitative survey data and in-depth qualitative interviews, collected before and after the onset of COVID-19, with over 3,000 Syrian refugees, stateless Palestinians and vulnerable Jordanians, living in camps, host communities and informal tented settlements. We utilize mixed-methods analysis combining multivariate regression with deductive qualitative tools to evaluate pandemic impacts and associated policy responses on adolescent wellbeing and mental health, at three and nine months after the pandemic onset. We also explore the role of support systems at individual, household, community, and policy levels.

### Findings

We find the pandemic has resulted in severe economic and service disruptions with far-reaching and heterogenous effects on adolescent wellbeing. Nine months into the pandemic, 19.3% of adolescents in the sample presented with symptoms of moderate-to severe

the UK Data Archive (http://doi.org/10.5255/UKDA-SN-8866-1).

**Funding:** Funding was received from the Research and Evaluation Division of the UK Foreign Commonwealth and Development Office (FCDO) for the Gender and Adolescence: Global Evidence (GAGE) longitudinal study. In addition, supplementary funding was provided by BMGF through the EMERGE PROJECT: OPP1163682 and INV018007 and Research Grants on Women, Victimization, and COVID-19 (# INV-003527). Dr Sarah Baird is the recipient of the latter. The funders had no role in study design, data collection and analysis, decision to publish, or preparation of the manuscript.

**Competing interests:** The authors have declared that no competing interests exist.

depression, with small signs of improvement (3.2 percentage points [pp], p<0.001). Two thirds of adolescents reported household stress had increased during the pandemic, especially for Syrian adolescents in host communities (10.7pp higher than any other group, p<0.001). Social connectedness was particularly low for girls, who were 13.4 percentage points (p<0.001) more likely than boys to have had no interaction with friends in the past 7 days. Adolescent programming shows signs of being protective, particularly for girls, who were 8.8 percentage points (p<0.01) more likely to have a trusted friend than their peers who were not participating in programming.

## Conclusions

Pre-existing social inequalities among refugee adolescents affected by forced displacement have been compounded during the COVID-19 pandemic, with related disruptions to services and social networks. To achieve Sustainable Development Goal targets to support healthy and empowered development in adolescence and early adulthood requires interventions that target the urgent needs of the most vulnerable adolescents while addressing population-level root causes and determinants of psychosocial wellbeing and resilience for all adolescent girls and boys.

## Introduction

The COVID-19 pandemic and associated risk-mitigation strategies have altered the social contexts in which adolescents in low- and middle-income countries (LMICs) live. These changes may have profound implications for adolescent capabilities and broader wellbeing, especially for the most vulnerable adolescents, including those living in poverty and adolescents affected by conflict and displacement. The pandemic has disrupted access to health services, through lower demand (due to fear of infection) and COVID-19-related health facility closures and reduced services [1, 2]. School closures have limited not only adolescents' learning, but also their peer interactions and access to school-based non-educational services. The economic impacts of the pandemic have been considerable and are expected to push 71 million people into extreme poverty [3]. These economic shocks have further implications for food security and nutrition, which are particularly important for adolescents [4]. Moreover, these multi-layered impacts may differentially affect refugee populations, as they already face challenges to socio-economic inclusion related to obtaining legal work permits, being more reliant on support from social safety nets and aid programs currently facing budget cuts, and are disproportionately likely to be employed in sectors that have been impacted by pandemic control measures [5, 6].

The need for improved outcomes for children and adolescents cannot be overstated if the international community is to achieve the Sustainable Development Goals (SDGs)—for adolescent sexual and reproductive health and mental health (SDG 3), education and learning (SDG 4), gender equality and the prevention of harmful practices such as child marriage (SDG 5), and reduced inequalities (SDG 10). Understanding the patterning of the impacts of the pandemic on adolescent lives, therefore, is critical to inform national post-pandemic response plans [7].

Although adolescents generally have a lower risk of mortality from COVID-19 than adults [8], the pandemic is having significant negative impacts on young people's mental health [9–

13]. In LMICs, young people are exhibiting symptoms of depression, anxiety, and post-traumatic stress—with girls, older adolescents, and those with pre-existing vulnerabilities most at risk [14–19]. Elevated levels of household stress as a result of economic constraints and limited privacy—especially in contexts of crowded living conditions—may be associated with increased risks of emotional, physical, and sexual violence in the home, especially for girls and young women [20–23]. These risks may be emerging in the context of already high levels of vulnerability to age- and gender-based forms of violence experienced by young people in forced displacement [6, 24]. However, while the World Bank and the United Nations Refugee Agency (UNHCR) report growing concerns about deepening vulnerabilities among refugee populations during the pandemic [25], little is known about its impacts on displaced populations, and how these impacts are compounded by gender- and age-related vulnerabilities [26, 27].

Now, more than 18 months since the pandemic's onset, its impacts on adolescent mental health are likely to persist, in light of risk-mitigation strategies that have disrupted adolescents' access to education and peer interactions—both of which are central to healthy adolescent development [12, 13, 28–33]. Given that most of the 307 million students experiencing school closures (as of December 2020) are located in the global South [34], the least advantaged young people are likely to bear the brunt of COVID-19's longer-term academic, social and mental health impacts. For girls, restrictive gender norms that limit their movement and access to communications technology even under normal circumstances mean that they may experience the worst effects of prolonged social isolation [23, 35]. Moreover, evidence suggests that girls from refugee communities may face even greater restrictions on account of concerns about sexual harassment and safety within host communities, camps and informal tented settlements (ITS) [36–38].

To mitigate the long-term negative impacts of the pandemic on adolescent wellbeing, strategies are needed to foster adolescent resilience. Adaptive and problem-focused coping strategies—including information-seeking, development of personal interests, maintenance of good habits (e.g., sleep, nutrition, and exercise), and volunteering to help others—have reduced adolescent distress globally [39–47]. Close relationships with parents, peers, and other trusted adults also are protective of adolescent wellbeing, and have been effective in conflict settings [48]. While social distancing measures have provided many families with time to strengthen parent-child relationships, evidence from high-income countries suggests that lockdowns have strained these relationships [49–51] and that close peer relationships are vital to support adolescents' resilience [33, 41, 43, 52, 53].

This analysis extends existing evidence to explore the direct effects of service closures and the associated economic downturn on adolescent girls and boys and their families from refugee and host communities in Jordan [54]. We also explore the indirect effects on adolescent social connectedness and resilience. We draw on longitudinal data collected from more than 3,300 adolescents, including 2,542 Syrian refugees, 259 stateless Palestinians, and 510 vulnerable Jordanians, living in diverse contexts in Jordan: camps, host communities, and informal tented settlements. (Although most Palestinian refugees have Jordanian citizenship, those formerly from the Gaza Strip remain stateless and highly vulnerable, given their exclusion from access to many labour-market opportunities and governmental social assistance [55]. The Palestinian sample focuses on stateless Palestinians from Jerash/Gaza refugee camp.).

Using mixed-methods data collected pre-pandemic (baseline conducted in 2018–2019) and at two time points after the pandemic's onset (May-July 2020 and November 2020-January 2021), we evaluate the impacts of the pandemic and associated policy responses on adolescent wellbeing and mental health. We also explore the role of support systems at the individual, household, community, and policy levels among communities affected by displacement. The

data allows us to explore whether the pandemic has exacerbated pre-existing social inequalities—including those between citizens and refugees, in-school and out-of-school adolescents, girls and boys, and adolescents from lower and higher asset decile cohorts—and the role of basic service and social protection provisioning in mitigating impacts for refugee and host-community populations in Jordan.

## Materials and methods

### Setting

Cases of COVID-19 were first detected in Jordan in March 2020 [56]. The government responded quickly and decisively with a strict lockdown, shutting all borders, and providing detailed daily televised briefings to the public. This response amounted to a score (in mid-March 2020) of 100 on the stringency index (a scale that takes on values of 0–100 with 100 being the strictest), and the index ranged from 50–80 in Jordan during the time of data collection [57]. The government also rapidly pivoted the educational system from in-person to virtual schooling (with schools reopening only briefly for a month in September before being closed again in response to escalating case numbers). For the first six months after the onset of the pandemic, Jordan's public health response was hailed as a success for containing viral transmission, with only 26 deaths and fewer than 4,000 cases (40.5 cumulative cases per 100,000) as of 17 September 2020 [56]. However, as the economic toll widened and deepened, with an estimated 5.5% contraction of the economy in 2020 [58], the government started to relax restrictions. The first cases of COVID-19 reported in the country's two main Syrian refugee camps—Azraq and Zaatari—were identified in September 2020 [59]. By mid-November 2020, the country had one of the highest per capita infection rates in the world, with almost 6,000 new infections a day, peaking at 77.8 per 100,000 on 18 November 2020 [56]. By June 2021, Jordan had recorded a total of more than 7,349 cumulative cases per 100,000 and 95.2 deaths per 100,000 [56].

Despite concerns about the vulnerability of refugee camps to the spread of COVID-19 [60, 61], rates of COVID-19 in Azraq and Zaatari refugee camps have generally remained below the rates seen in the Jordanian population overall, with a cumulative case rate of less than 4,000 per 100,000 across both camps as of August 2021 [62]. Even so, refugee populations in Jordan have experienced differential economic impacts due to the COVID-19 pandemic compared to Jordanian nationals; furthermore, a majority of Syrian refugees live outside of formal refugee camps, largely in host communities. According to a recent World Bank report, although a greater percentage of Jordanian households fell below the poverty line as a result of the COVID-19 pandemic than Syrian refugees living in Jordan, this differential is primarily because so many Syrian households already were below the poverty line before the pandemic began [63]. Refugees living in informal tented settlements (ITS) may be particularly vulnerable to the impact of the pandemic, given pre-existing poor access to health care and education, as well as challenges in securing safe water and sanitation facilities [64].

The government's current focus is on rolling out a vaccination campaign, which was launched in January 2021, and includes refugees and asylum-seekers, who are entitled to receive the vaccine free of charge [65]. By June 2021, more than 3 million vaccine doses had been administered [56], and a third of eligible refugees had been vaccinated in Jordan's refugee camps [66].

### Sample and study design

This article draws on data collected as part of ongoing longitudinal research in Jordan by the Gender and Adolescence: Global Evidence (GAGE) programme, which has been following

more than 3,000 adolescent boys and girls in camp and host communities since October 2018, including Syrian and Palestinian refugees, and vulnerable Jordanians. The Syrian refugee adolescents in the sample are predominantly from Daraa governorate in southwestern Syria, whose families fled to Jordan between 2012 and 2016. They are living either in Syrian refugee camps at Zaatari and Azraq, among host communities in Amman, Mafraq, and Irbid governorates, or in informal tented settlements. The Palestinian adolescents are stateless and live predominantly in Gaza/Jerash camp and surrounding communities in Jerash governorate. The Jordanian adolescents live in host communities in Amman, Mafraq, and Irbid governorates, and are from households that meet the United Nations (UN) Vulnerability Assessment Framework (VAF) criteria of multidimensional vulnerability. The framework encompasses expenditure and indebtedness, household structure including gender and dependency ratio, and type of coping strategies—for example, buying food on credit, accepting socially degrading or exploitative work, and reducing essential non-food expenditures [67].

The GAGE conceptual framework informs our analysis. Drawing on the work of Amartya Sen [68] and Martha Nussbaum [69], this framework adopts a capabilities approach to emphasize that investments in adolescents must support their development in multiple, interconnected domains of life. This investment includes support and services so that adolescents can navigate multidimensional physical, psycho-emotional, cognitive, and social changes that characterize the second decade of life, enabling them to achieve positive outcomes in key domains, namely: education and learning; bodily integrity and freedom from violence; health, nutrition, and sexual and reproductive health; voice and agency; psychosocial wellbeing; and economic empowerment [70]. In line with intersectional approaches, the framework pays particular attention to the context-appropriate types of support that young people need, and especially those who are socially disadvantaged (e.g., based on marital, disability or refugee status), to realize their full human capabilities [71].

## Quantitative sample and tools

Adolescents in the GAGE longitudinal sample, and their primary female caregivers, were surveyed initially between October 2018 and March 2019, when they were aged 10–12 years (the younger cohort) and 15–17 years (the older cohort). The sample includes vulnerable Jordanian, Syrian, and Palestinian adolescents living in camps (Azraq, Zaatari and Jerash/Gaza), host communities or informal tented settlements in five governorates of Jordan: Amman, Mafraq, Irbid, Jerash, and Zarqa. The full sample also includes subsamples of especially vulnerable adolescents, namely married girls and out-of-school adolescents. Adolescents were randomly sampled from databases of vulnerable adolescents maintained by UNHCR and the United Nations Children's Fund (UNICEF), with over-sampling of adolescents who had married before the age of 18 years, and some who married as young as 12 (including currently married, separated or divorced girls). For more details on the sampling strategy, see [72].

Building on the baseline data, two rounds of phone COVID-19-related quantitative surveys (COVID-R1 and COVID-R2) were undertaken between May 2020 and January 2021, again with the adolescents and their primary female caregivers. Because over 99% of households in the baseline sample had at least one working mobile phone, these phone surveys were conducted using tablets and computer-assisted telephone interviewing software through SurveyCTO. The questionnaire focused on the impacts of COVID-19 on GAGE's six capability domains (for full details see [73, 74]). We completed virtual interviews at COVID-R1 or COVID-R2 with 3,311 of the 4,025 adolescents who were surveyed in-person at baseline (an 82.3% follow-up rate). Phone survey data were collected with 3,020 adolescents in COVID-R1 (a 75% follow-up rate), which took place between May and July 2020; and with 2,909 adolescents in

COVID-R2 (a 72% follow-up rate), which took place between October 2020 and January 2021. Of these adolescents, 2,574 (64%) were interviewed in COVID-R1 and COVID-R2.

Our analysis focused on two main samples. The first was the 3,311 adolescents who were surveyed at either COVID-R1 or COVID-R2, and the second was the panel sample of 2,574 adolescents who were surveyed at COVID-R1 and COVID-R2. Table 1 provides details of the sample size by nationality across a variety of demographics.

The attrition rate of 18% (36% for the panel) was similar to those found in other phone surveys in LMICs during crises (including phone surveys undertaken during the COVID-19 pandemic), and it compared favourably to those with multiple survey rounds [75, 76]. Attrition in our sample—focusing on those we were unable to reach across either survey round—largely came from the respondent being unreachable in the time frame of our study (n = 560, 78.4% of attritors at COVID-R1 and n = 591, 82.8% of attritors at COVID-R2), as opposed to refusal (n = 150, 21.0% of attritors at COVID-R1 and n = 117, 16.4% of attritors at COVID-R2), or unable to be surveyed due to disability or death (n = 4 at COVID-R1 and n = 6 at COVID-R2, less than 1% of attritors at each round). The "unreachable" households included those whose contact phone numbers collected at baseline were no longer in use or no longer affiliated with the household, as well as phone lines with no answer after multiple attempts to make contact at different times of the day or different days of the week, with 15 calls conducted for households where the phone line was active but never answered, following best practices for phone surveys [77]. A small percentage of unreachable households were successfully contacted and consented to participate but could not be reached to conduct the actual survey. Thus, this

**Table 1. Descriptive statistics of GAGE Jordan quantitative sample, overall and by nationality.**

|  | Jordanian | Palestinian | Syrian | Total |
|---|---|---|---|---|
| Boys | 191 (120) | 107 (78) | 1305 (1025) | 1603 (1223) |
| Girls | 319 (212) | 152 (125) | 1237 (1014) | 1708 (1351) |
| Age 15–21 years* | 226 (127) | 99 (72) | 1314 (1024) | 1639 (1223) |
| Age 10–14 years** | 284 (205) | 160 (131) | 1228 (1015) | 1672 (1351) |
| Host communities | 504 (328) | 26 (18) | 1452 (1165) | 1982 (1511) |
| Informal tented settlements (ITSs) | 5 (3) | 0 (0) | 264 (225) | 269 (228) |
| Camp | 1 (1) | 233 (185) | 826 (649) | 1060 (835) |
| Pre-pandemic upper 50% assets | 408 (263) | 129 (99) | 919 (734) | 1456 (1096) |
| Pre-pandemic lower 50% assets | 102 (69) | 130 (104) | 1623 (1305) | 1855 (1478) |
| In-school pre-pandemic | 438 (305) | 222 (178) | 1760 (1467) | 2420 (1950) |
| Out-of-school pre-pandemic | 72 (27) | 37 (25) | 782 (572) | 891 (624) |
| Ever married, female adolescents (age 15 or older) | 33 (12) | 9 (8) | 191 (143) | 233 (163) |
| Total | 510 (332) | 259 (203) | 2542 (2039) | 3311 (2574) |

*Notes*: This table presents the demographic details of 3,311 Jordanian, Palestinian, and Syrian adolescents interviewed at least once during the COVID-19 pandemic (COVID-R1 or COVID-R2), as well as the demographic details of 2,574 adolescents in the panel sample interviewed at COVID-R1 and COVID-R2 (in parentheses). This table uses data on nationality, gender, age, location, and marital status, as reported in the most recently available survey (COVID-R1 or COVID-R2). Data on household assets is drawn from the baseline survey. An adolescent was considered to be enrolled in school in March 2020 if he or she reported formal school enrolment prior to the pandemic at either COVID-R1 or COVID-R2. Note that these samples exclude 8 Syrian adolescents interviewed at baseline and at least once during the COVID-19 pandemic who left Jordan after the baseline survey (2018–2019).

*Based on most recently available survey, 3 adolescents reported an age of "21" at their most recent survey during the COVID-19 pandemic. The majority of adolescents in the older cohort report an age of 15–20.

**Based on the most recently available survey, 1 adolescent reported an age of "10" at their most recent survey during the COVID-19 pandemic, and 18 adolescents reported an age of "11"; the vast majority of adolescents in the younger cohort range in age from 12 to 14.

sample should largely be considered a convenience sample of our original cohort that could be interviewed within our time and budget constraints.

Ultimately, the sample of 3,311 adolescents surveyed at either COVID-R1 or COVID-R2 appeared similar to the baseline sample on all characteristics, including wealth, formal school enrolment status, highest grade attended, age, gender, nationality, location, disability status, and marital status. No significant difference was apparent in the likelihood of being surveyed on any of these baseline characteristics, suggesting that attrition is largely "at random". In looking at the panel sample of 2,574 adolescents, we were significantly more likely to survey adolescents in both rounds from the younger cohort (ages 10–12 at baseline), refugee adolescents, and those living in informal tented settlements (S1 Table), but again, no difference in likelihood of being surveyed was apparent in both rounds based on baseline wealth, school enrolment status, highest grade attended, gender, disability status, or marital status. This limited differential attrition suggested that—if anything—the panel data sample was likely to be a slightly more vulnerable sample than the original baseline sample. Results should be interpreted with this in mind.

The quantitative analysis examined a large set of self-reported outcome measures from the three rounds of data collection that connected to the themes of disruption to services, psychosocial wellbeing, coping, and social connection. These variables are largely self-explanatory and are defined in the tables presented (see also details in S1 File). We also examined several indices that have been validated among adolescent populations in LMICs. First, to measure psychosocial wellbeing, we used the Patient Health Questionnaire 8 (PHQ-8), a brief screening questionnaire for depression [78]; and the Generalized Anxiety Disorder 7 (GAD-7) scale, a screening tool used for various anxiety disorders [79]. To measure coping, we used the Brief Resilient Coping Scale (BRCS), designed to capture the tendencies of individuals to cope with stress in a highly adaptive manner [80]. Finally, we used the Household Food Insecurity Access Scale (HFIAS) to measure food insecurity [81]. The three measures of wellbeing showed adequate reliability in our sample, with Cronbach's alphas of 0.79 in COVID-R1 and 0.83 in COVID-R2 for the PHQ-8, 0.86 in COVID-R2 for the GAD-7, and 0.70 for the BRCS in COVID-R2 [82, 83].

## Qualitative sample and tool

To complement and enrich the survey findings, we conducted qualitative research through telephone interviews between April and July 2020 with 104 girls and 74 boys who were also part of the quantitative sample, building on data with the same cohort of adolescents collected between September/October 2018 and 2019 (Table 2). The initial selection of adolescents for the qualitative subsample was purposive, based on ensuring inclusion of adolescent girls and

**Table 2. GAGE Jordan qualitative sample.**

|  | Host communities | Informal tented settlements | Camps | Total |
|---|---|---|---|---|
| Younger girls (10–14) | 14 | 9 | 13 | **36** |
| Younger boys (10–14) | 14 | 8 | 12 | **34** |
| Older girls (15–19) | 30 | 15 | 23 | **68** |
| Older boys (15–19) | 18 | 9 | 13 | **40** |
| In-school pre-pandemic | – | – | – | **93** |
| Married adolescent girls | – | – | – | **21** |
| Total adolescents | 76 | 41 | 61 | **178** |
| Total key informants | – | – | – | **65** |

boys from both age cohorts, and from different settings (camp, host community, and informal tented settlements), as well as the most socially disadvantaged adolescents (adolescents who were ever married, those with disabilities and those who were out of school). Researchers used tailored in-depth interview tools that focused on young people's short-term perceptions and experiences of the COVID-19 pandemic and associated service closures, including questions related to psycho-emotional coping repertoires and social connectedness with family and peers (for more details on these tools, see [84]).

We also conducted in-depth interviews with 65 key informants involved in service provision within community-based organizations (CBOs), non-governmental organizations (NGOs) and UN agencies. The key informants were selected based on a snowballing approach, whereby we sought service providers with community-specific and adolescent-specific knowledge to understand the context and programming dynamics that shape adolescent psychosocial wellbeing and resilience, both pre- and post-pandemic (see [84] for more details).

We selected researchers who had been involved in previous research with GAGE participants and, where possible, were of the same nationality as the interviewee. They carried out the telephone interviews in colloquial or modern standard Arabic as appropriate, drawing on virtual qualitative research guidance developed by GAGE [84].

## Data analysis

**Quantitative data analysis.**   Quantitative data was analyzed using Stata version 16.1. Using regression methods, we first provided baseline (pre-COVID-19) descriptive statistics of our sample of 3,311 adolescents surveyed at COVID-R1 or COVID-R2. Specifically, we provided the mean or proportion of the outcome under study across comparator groups of interest, and test for statistically significant differences in those means using a z-test in the case of two means (proportions) and an F-test of overall significance when comparing more than two groups. Our comparators of interest included: gender; baseline asset index categorized into deciles and then dichotomized into above or below median assets (constructed following the methods of Filmer and Pritchett [85]); location; refugee status; baseline schooling enrolment status (older adolescents); and marital status (older female adolescents).

Building on this bivariate analysis, we then conducted multivariate analysis to examine heterogeneity in impacts during COVID-19 across the same set of baseline characteristics described above. We estimated a linear probability model using the following equation:

$$Y_{i2} = \alpha + X'_{i1}\gamma + \varepsilon_{i2}, \tag{1}$$

where $Y_{i2}$ was the outcome of interest measured in two ways: (i) as a binary measure that took on a value of one if the adolescent experienced that outcome at either COVID-R1 or COVID-R2 (using the full sample of 3,311 adolescents); and (ii) as changes between COVID-R1 and COVID-R2 to document dynamics of the pandemic (using the panel sample of 2,574 adolescents). The vector $X'_{i1}$ included the baseline comparators of interest. We preferred the linear probability model for ease of interpretation, and the literature suggests that model performs as well as others [86, 87]. The linear probability model implies that impacts are measured in percentage point differences or risk differences. Note that results were qualitatively the same if we used a logit model. In addition, for binary variables, in the tables and figures we provided the proportion, while in the text we frequently converted to percentages for readability.

Finally, we explored whether adolescent-specific programming (one-stop centres for children and adolescents) acted as a protective factor. Makani (meaning 'my space' in Arabic) centres are a multi-donor funded initiative managed by UNICEF Jordan, reaching approximately 100,000 adolescents each year. Makani centres are implemented through CBOs, NGOs, and

the Ministry of Social Development in host communities, and by Syrian volunteers in refugee camps. One of their core aims is to reconnect refugees to formal schooling [88]. Given likely selection into Makani participation (with those who choose to participate being different from those who choose not to), we controlled for selection on observables using nearest-neighbour matching and five matches per observation. We matched participants to non-participants on a set of baseline variables—including asset decile, gender, age cohort, nationality, location, household size, whether the household is female headed, and if the household receives any aid —that are arguably exogenous to Makani participation. We used the 'teffects' command in Stata, which computes robust Abadie-Imbens standard errors [89]. All respondents found matches within the specified distance (using Mahalanobis distance for non-exact matches), and the matched standardized differences were all close to zero (largest is 0.02), while the variance ratio was close to 1 (ranges from 0.956 to 1.02). We required exact matches for gender, age cohort, nationality, and location.

When not looking at changes (which, by necessity, is restricted to the panel sample), our analysis focused on the sample of 3,311 adolescents that responded to either the COVID-R1 or COVID-R2 survey. This sample was preferred for two reasons: (i) it maximized the study's power; and (ii) there was no evidence of selection on observables. That said, we recognize there may be concerns that by combining COVID-R1 and COVID-R2 data, there may have been misclassification of outcomes due to missing data at one round. To address this concern, we re-ran all analysis restricted to the panel sample (results not shown, but available upon request), and results were qualitatively the same.

Throughout the manuscript we focus the written narrative on findings that were statistically significant at 0.05 significance or less, with all findings in the tables.

**Qualitative data analysis.** The qualitative interviews were transcribed, translated, and coded largely deductively according to a thematic coding book drawing on the capability domains covered in the tools, using the software package MAXQDA 12. Research teams held debriefing sessions (during and immediately after data collection) to discuss emerging findings and capture country-specific issues, and these inductively derived codes were added to the codebook accordingly. During qualitative data analysis, care was taken to identify themes that resonated beyond individuals and across the cohort or specific subgroups of adolescents within the cohort; the selected quotes are used to illustrate these insights. The quantitative and qualitative data was then triangulated to reveal both consistencies and inconsistencies across methods, with the qualitative data providing greater contextual nuance to the statistical findings.

## Ethics

Research ethics approvals were obtained through the Overseas Development Institute's Research Ethics Committee (02438), and the George Washington University Committee on Human Research Institutional Review Board (071721) and updated for the phone-based data collection following the onset of the Covid-19 pandemic. For research participants in refugee camps, we applied for and were granted permission from UNHCR's National Protection Working Group. For research participants in host communities, approval was granted by Jordan's Ministry of Interior, the Department of Statistics, and the Ministry of Education. Researchers were trained on how to interact appropriately with adolescents, including adolescents with different types of disadvantage (such as ever-married girls). Verbal consent was obtained from caregivers (typically the primary female caregiver) and adolescents over 18, and verbal assent to participate in the research was obtained from adolescents under 18 years old. Surveys were translated into modern standard Arabic and were piloted before being undertaken across the sample.

## Results

To explore the impact of the COVID-19 pandemic on adolescent health and wellbeing and explore the differences in pandemic effects based on pre-existing socioeconomic inequalities, we present our results in four sections. First, we explore pre-COVID-19 vulnerabilities to establish the existing inequalities in our sample prior to the pandemic onset. Then, we present findings on post-COVID-19 disrupted social contexts, describing service closures and livelihood restrictions reported by households in the sample. Next, we explore findings on COVID-19 and adolescent psychosocial wellbeing, focusing on mental health and adolescents' own concerns about their future during the pandemic. Finally, we examine challenges to coping and resilience under COVID-19, with a focus on both individual-level coping skills and resilience as well as social connectedness and support. Throughout each section, we highlight differences by gender, age, nationality, type of community, baseline assets, baseline school enrolment, and marital status and intertwine quantitative and qualitative findings on each of these topics.

### Pre-COVID-19 vulnerabilities

Our analysis begins with a brief overview of adolescents' (both age cohorts) wellbeing and access to support structures *before the pandemic* (in 2019), and the patterning of pre-existing vulnerabilities (Table 3). Looking first at broad characteristics of the sample, adolescents from below-median asset households, across nationalities, are less likely to be enrolled in formal school than adolescents from above-median asset households (68.4% vs. 83.4%, p<0.001), and more likely to be married (5.7% vs. 3.3%, p<0.010). Syrian refugee adolescents are less likely to be enrolled in school (70.6%) than their Jordanian (90.2%) and Palestinian (88.4%) peers (p<0.001), with those Syrian refugees living in informal tented settlements much less likely to be enrolled (38.3%). Syrian refugee adolescents are also likely to have fewer assets and are more likely to be married than Palestinian and Jordanian adolescents (5.3% vs. 3.3% for Jordanians and 0.8% for Palestinians, p<0.001).

While Syrian adolescents are no more likely to be food insecure or hungry than adolescents of other nationalities, there is substantial variation by location. Syrians who live in host communities and ITS are more likely to be food insecure than those in camps (58.5% host vs. 56.3% ITS vs. 38.8% camp, p<0.001) and more likely to have been hungry in the past four weeks (19.3% host vs. 22.7% ITS vs. 9.7% camp, p<0.001). Among Syrian households, those living in ITSs are also the least likely to receive food or cash aid (89.7% in ITS vs. 93.1% host vs. 98.6% camp, p<0.001) and to attend Makani adolescent programming (33% in ITS vs. 38.6% host vs. 74.1% camp, p<0.001). Despite these disparities, adolescents in ITS communities also are the least likely to report experiencing household violence in the last 12 months among Syrian refugees by location (38.8% ITS vs. 40.5% camp vs. 51.3% host, p<0.001) and also most likely to report having a trusted friend (76.1% ITS vs. 74.5% camp vs. 69.4% host, p = 0.008).

Girls are more restricted in their mobility (66.7% leave their house daily compared to 86.9% of boys, p<0.001). Significantly more restrictions are placed on married girls than on unmarried girls, with only 26% leaving their house daily and relatively fewer having a trusted friend (58.7% vs. 74.2%, p<0.001). Boys, however, are slightly more likely to experience violence in the household (52.5% vs. 46.7% of girls, p<0.001). In terms of access to aid, 74.3% of households receive some type of cash or food support, with this aid largely concentrated among Syrians (94.5% of Syrians in the sample reported receiving support).

When looking at adolescents who have received support from the UNICEF Makani programme, rates are highest among Palestinians (68.0%), followed by Jordanians (51.4%), and

**Table 3. Baseline characteristics (2018–2019) of the GAGE Jordan quantitative sample (adolescents interviewed at baseline and COVID-R1 or COVID-R2).**

Panel A. Overall and by baseline (pre-COVID-19) wealth, nationality, and location of residence (n = 3,311)

| Measure | Overall mean (sd) | Wealth | | | Nationality | | | | Location (Syrian only) | | | |
|---|---|---|---|---|---|---|---|---|---|---|---|---|
| | | Below median assets mean (sd) | Above median assets mean (sd) | Effect size (se) | Jordanian mean (sd) | Syrian mean (sd) | Palestinian mean (sd) | Overall p-value | Syrian-host mean (sd) | Syrian-ITS mean (sd) | Syrian-camp mean (sd) | Overall p-value |
| = 1 if adolescent experienced violence at home in past 12 months, including being yelled at or hit by parent or male in the household | 0.495 (0.500) | 0.453 (0.498) | 0.548 (0.498) | 0.095*** (0.017) | 0.620 (0.486) | 0.465 (0.499) | 0.544 (0.499) | <0.001 | 0.513 (0.500) | 0.388 (0.488) | 0.405 (0.491) | <0.001 |
| = 1 if adolescent ever attended Makani (adolescent one-stop programming) | 0.513 (0.500) | 0.475 (0.500) | 0.560 (0.497) | 0.085*** (0.017) | 0.514 (0.500) | 0.496 (0.500) | 0.680 (0.468) | <0.001 | 0.386 (0.487) | 0.333 (0.472) | 0.741 (0.438) | <0.001 |
| = 1 if household currently receives food or cash aid | 0.743 (0.437) | 0.832 (0.374) | 0.625 (0.484) | 0.207*** (0.015) | 0.029 (0.169) | 0.945 (0.228) | 0.194 (0.396) | <0.001 | 0.931 (0.254) | 0.897 (0.305) | 0.986 (0.117) | <0.001 |
| = 1 if adolescent leaves home daily | 0.765 (0.424) | 0.748 (0.434) | 0.786 (0.410) | 0.037* (0.015) | 0.737 (0.441) | 0.769 (0.422) | 0.782 (0.414) | 0.249 | 0.723 (0.448) | 0.712 (0.454) | 0.867 (0.340) | <0.001 |
| = 1 if adolescent has a friend that he/she trusts | 0.712 (0.453) | 0.702 (0.457) | 0.725 (0.447) | 0.023 (0.016) | 0.729 (0.445) | 0.717 (0.450) | 0.633 (0.483) | 0.011 | 0.694 (0.461) | 0.761 (0.427) | 0.745 (0.436) | 0.008 |
| = 1 if household is severely food insecure based on Food Insecurity Access Scale (FIAS) | 0.516 (0.500) | 0.575 (0.495) | 0.440 (0.497) | -0.135*** (0.018) | 0.497 (0.500) | 0.520 (0.500) | 0.520 (0.501) | 0.641 | 0.585 (0.493) | 0.563 (0.497) | 0.388 (0.488) | <0.001 |
| = 1 if adolescent self-reports being hungry in the prior 4 weeks | 0.166 (0.372) | 0.192 (0.394) | 0.134 (0.341) | -0.057*** (0.013) | 0.173 (0.379) | 0.165 (0.372) | 0.163 (0.370) | 0.898 | 0.193 (0.395) | 0.227 (0.420) | 0.097 (0.296) | <0.001 |
| = 1 if adolescent is enrolled in school | 0.751 (0.433) | 0.685 (0.465) | 0.834 (0.372) | 0.15*** (0.015) | 0.902 (0.298) | 0.707 (0.455) | 0.884 (0.321) | <0.001 | 0.738 (0.440) | 0.383 (0.487) | 0.755 (0.430) | <0.001 |
| Decile of assets (1–10) | 5.167 (2.935) | 2.872 (1.391) | 8.091 (1.364) | 5.22*** (0.048) | 7.633 (2.579) | 4.633 (2.729) | 5.548 (3.016) | <0.001 | 4.393 (2.651) | 3.292 (2.196) | 5.485 (2.763) | <0.001 |
| = 1 if adolescent has ever been married | 0.047 (0.211) | 0.057 (0.232) | 0.033 (0.179) | -0.024** (0.007) | 0.033 (0.180) | 0.053 (0.224) | 0.008 (0.088) | 0.001 | 0.058 (0.234) | 0.045 (0.209) | 0.047 (0.212) | 0.466 |
| Sample size | 3311 | 1456 | 1855 | | 510 | 2542 | 259 | | 1452 | 264 | 826 | |

Panel B: Overall and by adolescent gender and adolescent age cohort, as well as by school enrollment status (among adolescents age 15 and older), and marital status (among female adolescents age 15 and older)

| Measures | Overall Mean (sd) | Gender | | | Cohort | | | Baseline School Enrollment (age 15 and older only) | | | Ever Married (girls age 15 and older only) | | |
|---|---|---|---|---|---|---|---|---|---|---|---|---|---|
| | | Boy Mean (sd) | Girl Mean (sd) | Effect Size (se) | Younger Mean (sd) | Older Mean (sd) | Effect Size (se) | Out of School Mean (sd) | In School Mean (sd) | Effect Size (se) | Never Mean (sd) | Ever Mean (sd) | Effect Size (se) |
| = 1 if adolescent experienced violence at home in last 12 months, including being yelled at or hit by parent or male in the household | 0.495 (0.500) | 0.525 (0.500) | 0.467 (0.499) | -0.058*** (0.009) | 0.511 (0.500) | 0.479 (0.500) | 0.032* (0.010) | 0.401 (0.490) | 0.541 (0.499) | 0.139*** (0.025) | 0.514 (0.500) | 0.260 (0.440) | -0.254*** (0.044) |
| = 1 if adolescent ever attended Makani (adolescent one-stop programming) | 0.513 (0.500) | 0.515 (0.500) | 0.511 (0.500) | -0.005 (0.027) | 0.595 (0.491) | 0.429 (0.495) | 0.166* (0.045) | 0.299 (0.458) | 0.532 (0.499) | 0.233*** (0.024) | 0.458 (0.499) | 0.200 (0.401) | -0.258*** (0.043) |

*(Continued)*

| | | | | | | | | | | | | |
|---|---|---|---|---|---|---|---|---|---|---|---|---|
| = 1 if household currently receives food or cash aid from any of the following programs: WFP, UNHCR, UNRWA, or Hajati | 0.743 | 0.788 | 0.700 | -0.088** | 0.780 | 0.706 | -0.074 | 0.845 | 0.729 | -0.116*** | 0.718 | 0.750 | 0.032 |
| | (0.437) | (0.409) | (0.458) | (0.020) | (0.414) | (0.456) | (0.034) | (0.363) | (0.445) | (0.021) | (0.450) | (0.434) | (0.041) |
| = 1 if adolescent leaves home daily | 0.765 | 0.869 | 0.667 | -0.202*** | 0.715 | 0.814 | 0.099* | 0.568 | 0.831 | 0.263*** | 0.619 | 0.260 | -0.359*** |
| | (0.424) | (0.338) | (0.471) | (0.031) | (0.452) | (0.389) | (0.031) | (0.496) | (0.375) | (0.022) | (0.486) | (0.440) | (0.043) |
| = 1 if adolescent has a friend that he/she trusts | 0.712 | 0.729 | 0.697 | -0.033 | 0.736 | 0.689 | -0.047 | 0.674 | 0.786 | 0.112*** | 0.742 | 0.587 | -0.155*** |
| | (0.453) | (0.444) | (0.460) | (0.019) | (0.441) | (0.463) | (0.030) | (0.469) | (0.410) | (0.022) | (0.438) | (0.494) | (0.040) |
| = 1 if household is severely food insecure based on Food Insecurity Access Scale (FIAS) | 0.516 | 0.521 | 0.512 | -0.008 | 0.503 | 0.529 | 0.026 | 0.523 | 0.487 | -0.036 | 0.519 | 0.555 | 0.036 |
| | (0.500) | (0.500) | (0.500) | (0.017) | (0.500) | (0.499) | (0.019) | (0.500) | (0.500) | (0.025) | (0.500) | (0.499) | (0.046) |
| = 1 if adolescent self-reports being hungry in the prior four weeks | 0.166 | 0.167 | 0.166 | -0.001 | 0.147 | 0.185 | 0.038 | 0.162 | 0.136 | -0.026 | 0.152 | 0.200 | 0.048 |
| | (0.372) | (0.373) | (0.372) | (0.013) | (0.355) | (0.388) | (0.022) | (0.369) | (0.343) | (0.018) | (0.360) | (0.401) | (0.033) |
| = 1 if adolescent is enrolled in school | 0.751 | 0.740 | 0.761 | 0.021 | 0.559 | 0.938 | 0.390*** | 0.000 | 1.000 | 1.000*** | 0.672 | 0.060 | -0.612*** |
| | (0.433) | (0.439) | (0.427) | (0.039) | (0.497) | (0.241) | (0.031) | (0) | (0) | (0) | (0.470) | (0.238) | (0.039) |
| Decile of Assets (1–10) | 5.167 | 5.097 | 5.233 | 0.136 | 5.198 | 5.136 | -0.062 | 4.393 | 5.834 | 1.441*** | 5.394 | 4.373 | -1.021*** |
| | (2.935) | (2.886) | (2.980) | (0.117) | (2.930) | (2.94) | (0.153) | (2.721) | (2.934) | (0.141) | (2.941) | (2.884) | (0.264) |
| = 1 if adolescent has ever been married | 0.047 | 0.002 | 0.088 | 0.087* | 0.093 | 0.001 | -0.093* | 0.199 | 0.010 | -0.189*** | 0.000 | 1.000 | 1.000*** |
| | (0.211) | (0.043) | (0.284) | (0.032) | (0.291) | (0.024) | (0.029) | (0.400) | (0.099) | (0.014) | (0) | (0) | (0) |
| Sample Size | 3311 | 1603 | 1708 | | 1639 | 1672 | | 723 | 916 | | 698 | 150 | |

*Notes:* * indicated p<0.05

** p<0.01, and

*** p<0.001. Table presents means (proportions) of key variables of interest collected as part of the GAGE baseline survey in Jordan that took place from October 2018-March 2019 with adolescents aged 10–17 years. The sample presented here includes results for 3,311 Jordanian, Syrian, and Palestinian adolescents interviewed at baseline and during the COVID-19 pandemic, either at COVID-R1, COVID-R2, or both. The table provides the overall mean and standard deviation, and differences in mean by wealth, nationality, and location of residence, for Syrian adolescents only (n = 2,542). For comparisons of baseline wealth, the table presents the effect size, defined here as the coefficient for baseline wealth using a linear regression, and standard error. For binary measures, the coefficient can be interpreted as "percentage point difference," or risk difference, in the measure of interest for adolescents in households with above-median assets compared to those with below-median assets. For comparisons of nationality and location for Syrian refugees, the table presents the p-value for the results of the joint F-test.

*Notes*

* indicated p<0.05

** p<0.01, and

*** p<0.001. Table presents means of key variables of interest collected as part of the GAGE baseline survey in Jordan that took place from October 2018-March 2019 with adolescents aged 10–17. The sample presented here includes results for 3,311 Jordanian, Syrian, and Palestinian adolescents interviewed at baseline and during the COVID-19 pandemic, either at COVID-R1, COVID-R2, or both. The table provides the overall mean and standard deviation, and differences in mean by gender, cohort (age 10–14 in the younger cohort and age 15 and older in the older cohort), baseline formal school enrolment (for adolescents age 15 or older, n = 848). For comparisons of gender, age cohort, school enrolment status, and marital status (adolescent girls age 15 and older, n = 1639) and marital status, the table presents the effect size, defined here as the coefficient for female gender, baseline school enrolment, and being ever-married using a linear regression, and standard error. For binary measures, the coefficient can be interpreted as "percentage point difference," or risk difference, in the measure of interest for adolescents who are girls, younger cohort, in-school, or ever married, compared to those who are not.

Syrians (49.6%)—again, with significantly lower rates among Syrians in host communities (38.6%) and those living in ITSs (33.3%). Adolescents from poorer households also have substantially less access to Makani (Table 3, Panel A). Almost all the adolescents who are married are girls (there are only three married boys in our sample).

Thus, before the onset of COVID-19, boys and girls faced varied disadvantages, with married girls being especially vulnerable on the characteristics presented in Table 3. Syrian refugees—especially those in ITSs and host communities—fared considerably worse than Jordanian nationals across most outcomes. Similarly, adolescents who were out of school and those in more economically vulnerable households faced multiple deprivations.

## Post-COVID-19 disrupted social contexts—service closures and livelihood restrictions

We now turn to findings of similar measures at the onset of the pandemic (COVID-R1) and nine months into the pandemic (COVID-R2). Findings from the COVID-R1 and COVID-R2 data underscore that the impacts of the pandemic in terms of household-level economic shocks, school closures, and disruptions to basic services have significantly affected adolescents' lives. At the onset of the pandemic, 73% of adolescents in the sample (or 76% of adolescents in the panel sample surveyed at both rounds) were enrolled in formal schools in Jordan, which closed in March 2020 and subsequently transitioned to virtual learning (apart from a two-week period in September 2020) until reopening for in-person schooling in September 2021. Alongside school closures, household-level livelihood shocks and closure of basic services have substantially disrupted the social contexts in which adolescents live (Table 4, Panel A). Most households (75.9%) lost some employment during the pandemic, with 83.0% reporting losing some income, and 65.2% unable to buy essential food items in the past week. This disruption has been especially acute for the most vulnerable adolescents, including those from poorer households, older adolescents who were out of school pre-pandemic, older adolescent girls who were ever married, and refugees (particularly Syrians). Within the Syrian population, impacts were more pronounced for adolescents living in host communities and ITSs compared to those living in camps.

While there were some small signs of recovery in terms of livelihoods by the time of the second survey round (October 2020 to January 2021), the negative shock has persisted throughout the pandemic (Table 4, Panel B). Significant improvements are observed for some outcomes for the panel sample interviewed at both rounds, but they are generally small in magnitude: a 6.7 percentage point (pp) reduction in lost employment (but still 59.0% of households reporting lost jobs); a 3.9pp reduction in household food insecurity (with 40.2% remaining food insecure); and a 4.6pp reduction in the adolescent reporting going hungry in the past four weeks (with 25.6% remaining hungry). No change has been observed in the proportion reporting lost income. Further, the second round of the survey saw an 11.4pp increase in households reporting that they were unable to buy essential food items in the past week (a total of 55.3% of the panel households report this at COVID-R2).

## COVID-19 and adolescent psychosocial wellbeing

While many adolescents are coping during the pandemic, there are observable signs of anxiety, fear, and depression, particularly in certain subgroups (Table 5, Panel A). Quantitatively, 19.3% of adolescents in the sample present with symptoms suggestive of moderate-to-severe depression at either COVID-R1 or COVID-R2, as measured by the PHQ-8, with lower rates among younger adolescents and Syrians living in camps. This percentage is lower than findings from other COVID-19 studies—perhaps because stress has been normalized for these

**Table 4. Disrupted services and economic impacts of COVID-19 pandemic.**

Panel A. Outcomes reported at COVID-R1 and/or COVID-R2 (among 3,311 adolescents interviewed at either round)

| Covariates | Household lost any employment due to pandemic | Household lost any income due to pandemic | Household unable to buy essential food items in prior week | Household severely food insecure (FIAS) | Adolescent unable to access healthcare due to pandemic (Covid-R2 only; among those who needed healthcare only) | Adolescent self-reported hunger in prior 4 weeks |
|---|---|---|---|---|---|---|
| = 1 if household had above-median assets at baseline | 0.019 | 0.024 | -0.070*** | -0.099*** | -0.031 | -0.101*** |
|  | (0.016) | (0.014) | (0.018) | (0.019) | (0.027) | (0.018) |
| = 1 if household is Syrian and lives in host community | 0.141*** | 0.061** | 0.125*** | 0.174*** | 0.019 | 0.097*** |
|  | (0.025) | (0.021) | (0.027) | (0.027) | (0.037) | (0.026) |
| = 1 if household is Syrian and lives in ITS | 0.093** | 0.012 | 0.040 | 0.147*** | 0.035 | 0.061 |
|  | (0.036) | (0.032) | (0.040) | (0.040) | (0.059) | (0.040) |
| = 1 if household is Syrian and lives in camp | 0.055* | -0.014 | -0.024 | 0.033 | 0.069 | -0.043 |
|  | (0.027) | (0.023) | (0.029) | (0.029) | (0.043) | (0.026) |
| = 1 if household is Palestinian | 0.157*** | 0.083** | 0.092* | -0.028 | 0.034 | -0.005 |
|  | (0.033) | (0.028) | (0.038) | (0.039) | (0.060) | (0.036) |
| = 1 if adolescent girl | -0.036* | -0.018 | -0.035* | -0.019 | -0.019 | -0.021 |
|  | (0.016) | (0.014) | (0.017) | (0.018) | (0.027) | (0.017) |
| = 1 if adolescent was enrolled in formal school immediately before pandemic | -0.030 | -0.004 | -0.013 | 0.008 | 0.073* | -0.005 |
|  | (0.021) | (0.019) | (0.023) | (0.024) | (0.036) | (0.023) |
| = 1 if adolescent was ever married (at any survey round) | 0.082** | 0.062* | -0.027 | 0.014 | -0.004 | 0.078* |
|  | (0.029) | (0.025) | (0.036) | (0.037) | (0.046) | (0.036) |
| = 1 if adolescent is in younger cohort (10–14 years) | -0.003 | -0.010 | -0.014 | 0.022 | -0.031 | 0.103** |
|  | (0.018) | (0.016) | (0.019) | (0.020) | (0.030) | (0.019) |
| Sample mean (sd) | 0.759 | 0.830 | 0.652 | 0.558 | 0.225 | 0.373 |
|  | (0.428) | (0.375) | (0.477) | (0.497) | (0.418) | (0.484) |
| Observations | 3,230 | 3,242 | 3,217 | 3,242 | 1,125 | 3,309 |

Panel B. Change in outcomes reported in COVID-R1 v. COVID-R2 (among 2,574 adolescents interviewed in both rounds)

| Covariates | Change in household lost employment | Change in household lost income | Change in household ability to buy essential food items | Change in severe food insecurity | Change in adolescent self-reported hunger in prior 4 weeks |
|---|---|---|---|---|---|
| = 1 if household had above-median assets at baseline | 0.017 | 0.021 | -0.053 | 0.062* | 0.000 |
|  | (0.025) | (0.024) | (0.029) | (0.026) | (0.021) |
| = 1 if household is Syrian and lives in host community | 0.039 | -0.014 | -0.002 | -0.057 | 0.034 |
|  | (0.038) | (0.038) | (0.041) | (0.037) | (0.032) |
| = 1 if household is Syrian and lives in ITS | -0.032 | 0.040 | 0.037 | 0.001 | 0.028 |
|  | (0.051) | (0.055) | (0.063) | (0.057) | (0.051) |
| = 1 if household is Syrian and lives in camp | 0.101* | 0.035 | -0.004 | -0.118** | -0.002 |
|  | (0.040) | (0.041) | (0.043) | (0.038) | (0.032) |
| = 1 if household is Palestinian | -0.032 | 0.015 | -0.117 | 0.014 | 0.089* |
|  | (0.056) | (0.049) | (0.060) | (0.051) | (0.044) |

(Continued)

| | | | | | |
|---|---|---|---|---|---|
| = 1 if adolescent girl | 0.018 | 0.010 | -0.008 | -0.018 | 0.006 |
| | (0.024) | (0.023) | (0.027) | (0.024) | (0.020) |
| = 1 if adolescent was enrolled in formal school immediately before pandemic | -0.013 | 0.019 | 0.049 | 0.071* | -0.005 |
| | (0.032) | (0.030) | (0.041) | (0.034) | (0.029) |
| = 1 if adolescent was ever married (at any survey round) | -0.105 | 0.019 | -0.041 | 0.073 | 0.005 |
| | (0.053) | (0.054) | (0.066) | (0.054) | (0.047) |
| = 1 if adolescent is in younger cohort (10–14 years) | -0.025 | -0.015 | 0.026 | 0.012 | 0.009 |
| | (0.026) | (0.025) | (0.031) | (0.027) | (0.022) |
| Panel mean for outcome at COVID-R2 (sd) | 0.590 | 0.711 | 0.553 | 0.402 | 0.256 |
| | (0.492) | (0.453) | (0.497) | (0.490) | (0.436) |
| Panel mean change between rounds (sd) | -0.067*** | 0.007 | 0.114*** | -0.039** | -0.046*** |
| | (0.553) | (0.529) | (0.625) | (0.569) | (0.495) |
| Observations | 2328 | 2356 | 2183 | 2352 | 2569 |

*Notes*: Standard errors of covariates in parentheses

* p<0.05

** p<0.01

*** p<0.001. This table presents the results of a linear multivariate regression of the outcomes of interest reported by interviewed adolescents and households at COVID-R1 and/or COVID-R2 regressed on the listed covariates. The outcomes "Lost any income," "Lost any employment," "Unable to buy essential food items in the past week," and "Household severely food insecure" are drawn from the adult female survey, which is available for 98% of adolescents who were interviewed at either COVID-R1 or COVID-R2 (adult female survey n = 3,246). For each outcome, adolescents (or adult female caregivers) who refused a question or responded "Don't know" are excluded from the sample for that outcome, resulting in the sample size range of n = 3,217 to n = 3,242 for questions answered by the adult female respondent and a sample size of n = 3,309 for the outcome on adolescent-reported hunger, asked to all adolescents at both survey rounds. Note that the outcome "Adolescent was unable to access healthcare" was only asked at COVID-R2 (n = 2,886); furthermore, this sample is restricted to those adolescents who reported needing to see a doctor or healthcare provider or needing to access medication in the time since March 2020 (n = 1,125).

Notes: Standard errors of covariates in parentheses

* p<0.05

** p<0.01

*** p<0.001. This panel presents the results of a linear multivariate regression of the change in key outcomes reported by adolescents and households interviewed at both COVID-R1 and COVID-R2) on the listed covariate. Note that changes in the outcomes "Lost any income," "Lost any employment," "Unable to buy essential food items in the past week," and "Household severely food insecure" are drawn from the adult female survey, which is available at both COVID-R1 and COVID-R2 for 92% of adolescents who were also interviewed at both COVID-R1 and COVID-R2 (adult female survey n = 2,361 for this panel). For each outcome, adolescents (or adult female caregivers) who refused a question or responded "Don't know" are excluded from the sample for that outcome, resulting in a sample size range of n = 2,183 to n = 2,356 for questions answered by the adult female respondent and a sample size of 2,569 for the question on adolescent-reported hunger in the past 4 weeks, asked to all adolescent at both survey rounds.

**Table 5. Psychosocial wellbeing.**

Panel A. Outcomes reported at COVID-R1 and/or COVID-R2 (among 3,311 adolescents interviewed at either round)

| Covariates | PHQ-8 results indicate symptoms of moderate-severe depressive (≥ 10) | Adolescent believes people in community are more anxious/depressed (COVID-R2 and age 15 + only) | Adolescent thinks there is an increase in thoughts of self-harm in community (COVID-R2 and age 15 + only) | GAD-7 results indicate for moderate-severe anxiety (≥ 10, COVID-R2 only) | Adolescent is fearful or embarrassed to ask family for menstrual hygiene management support (COVID-R2 only) | Adolescent worries about marrying earlier due to pandemic | Adolescent agrees pressure to marry decreased due to pandemic |
|---|---|---|---|---|---|---|---|
| = 1 if household had above-median assets at baseline | -0.014 | -0.013 | 0.035 | -0.000 | -0.019 | 0.002 | 0.012 |
|  | (0.015) | (0.022) | (0.026) | (0.013) | (0.029) | (0.014) | (0.017) |
| = 1 if household is Syrian and lives in host community | 0.027 | -0.028 | -0.147*** | 0.001 | 0.029 | 0.043* | 0.042 |
|  | (0.022) | (0.030) | (0.045) | (0.021) | (0.041) | (0.019) | (0.023) |
| = 1 if household is Syrian and lives in ITS | -0.035 | -0.111* | -0.264*** | -0.054 | 0.062 | 0.114*** | 0.089* |
|  | (0.032) | (0.051) | (0.057) | (0.028) | (0.063) | (0.034) | (0.037) |
| = 1 if household is Syrian and lives in camp | -0.055* | -0.077* | -0.209*** | -0.044* | -0.095* | 0.039 | 0.080** |
|  | (0.022) | (0.033) | (0.046) | (0.021) | (0.045) | (0.020) | (0.024) |
| = 1 if household is Palestinian | -0.024 | -0.049 | -0.097 | -0.043 | -0.032 | 0.077** | 0.048 |
|  | (0.030) | (0.049) | (0.066) | (0.027) | (0.061) | (0.029) | (0.032) |
| = 1 if adolescent girl | 0.008 | 0.047* | 0.035 | 0.045*** | – | -0.009 | -0.066*** |
|  | (0.014) | (0.021) | (0.026) | (0.013) | – | (0.014) | (0.016) |
| = 1 if adolescent was enrolled in formal school immediately before pandemic | 0.023 | 0.034 | -0.003 | 0.005 | 0.004 | -0.064** | -0.082*** |
|  | (0.020) | (0.022) | (0.027) | (0.018) | (0.041) | (0.020) | (0.023) |
| = 1 if adolescent was ever married (at any survey round) | -0.007 | -0.048 | -0.008 | -0.015 | -0.107* | – | – |
|  | (0.031) | (0.032) | (0.038) | (0.029) | (0.047) | – | – |
| = 1 if adolescent is in younger cohort (10–14 years) | -0.080*** | -0.134 | -0.004 | -0.045** | 0.030 | -0.041** | -0.093*** |
|  | (0.016) | (0.174) | (0.182) | (0.015) | (0.032) | (0.015) | (0.017) |
| Sample mean (sd) | 0.193 | 0.831 | 0.279 | 0.124 | 0.616 | 0.165 | 0.248 |
|  | (0.395) | (0.375) | (0.449) | (0.329) | (0.487) | (0.372) | (0.432) |
| Observations | 3307 | 1399 | 1381 | 2882 | 1270 | 3053 | 3049 |

Panel B. Change in outcomes reported in COVID-R1 vs. COVID-R2 (among adolescents 2,574 interviewed both rounds)

| Covariates | Change in rate of PHQ-8 score symptoms of depression (≥ 10) | Change in adolescents worrying about early marriage due to the pandemic | Change in adolescents reporting decreased pressure to marry due to the pandemic |
|---|---|---|---|
| = 1 if household had above-median assets at baseline | -0.014 | 0.035* | 0.006 |
|  | (0.018) | (0.018) | (0.022) |
| = 1 if household is Syrian and lives in host community | -0.019 | -0.000 | 0.009 |
|  | (0.028) | (0.024) | (0.031) |
| = 1 if household is Syrian and lives in ITS | -0.056 | 0.043 | -0.034 |
|  | (0.039) | (0.041) | (0.048) |

| | | | |
|---|---|---|---|
| = 1 if household is Syrian and lives in camp | -0.031 | 0.030 | 0.024 |
| | (0.028) | (0.025) | (0.033) |
| = 1 if household is Palestinian | -0.031 | 0.001 | -0.031 |
| | (0.036) | (0.035) | (0.041) |
| = 1 if adolescent girl | 0.028 | 0.011 | 0.018 |
| | (0.017) | (0.016) | (0.020) |
| = 1 if adolescent was enrolled in formal school immediately before pandemic | -0.054* | 0.000 | -0.095** |
| | (0.025) | (0.026) | (0.033) |
| = 1 if adolescent was ever married (at any survey round) | -0.025 | – | – |
| | (0.038) | – | – |
| = 1 if adolescent is in younger cohort (10–14 years) | 0.072*** | 0.036* | 0.038 |
| | (0.019) | (0.018) | (0.023) |
| Panel mean for outcome at COVID-R2 (sd) | 0.110 | 0.116 | 0.147 |
| | (0.313) | (0.321) | (0.355) |
| Panel mean change between rounds (sd) | -0.032*** | 0.020* | -0.008 |
| | (0.407) | (0.397) | (0.488) |
| Observations | 2565 | 2358 | 2347 |

Notes: Standard errors of covariates in parentheses

\* p<0.05

\*\* p<0.01

\*\*\* p<0.001. This table presents the results of a linear multivariate regression of the outcomes of interest reported by interviewed adolescents and households at COVID-R1 and/or COVID-R2 regressed on the listed covariates. For each outcome, adolescents who refused a question or responded "Don't know" are excluded from the sample for that outcome. The outcome "PHQ-8 results in range for moderate/severe depression" has a sample size of n = 3,307 after excluding refusals on any of the instrument's 8 questions. Note that the outcomes "Adolescent believes people in the community are becoming anxious/depressed due to COVID-19" and "Adolescent believes there is an increase in thoughts of self-harm in community due to COVID-19" are restricted to adolescents age 15 or older, and are only asked at COVID-R2 (n = 1,381 to n = 1,399 after excluding refusals). The outcome "GAD-7 results in range for moderate/severe anxiety" was only administered at COVID-R2, and the sample is restricted to those interviewed at this round who provided answers to all 7 questions in the scale (n = 2,882 after excluding refusals). The outcome "Adolescent is fearful or embarrassed to ask family for menstrual hygiene management" was only asked at COVID-R2 and is restricted to girls who reported reaching menarche who were interviewed at this round (n = 1,270 after excluding refusals). The outcomes "Adolescent worries about marrying earlier due to the pandemic" and "Adolescent agrees the pressure to marry has decreased due to the pandemic" are only asked to adolescents who are not married at any round (n = 3,049 to n = 3,053 after excluding refusals).

Notes: Standard errors of covariates in parentheses

\* p<0.05

\*\* p<0.01

\*\*\* p<0.001. This panel presents the results of a linear multivariate regression of the change in key outcomes reported by adolescents and households interviewed at both COVID-R1 and COVID-R2 on the listed covariates. For each outcome, adolescents who refused a question or responded "Don't know" are excluded from the sample for that outcome. Note that changes in the outcome related to PHQ-8 results are restricted to adolescents interviewed at both rounds (COVID-R1 and COVID-R2) who completed all 8 questions in the scale each time (n = 2,565 after excluding refusals). Changes in the outcomes "Adolescent worries about marrying earlier due to the pandemic" and "Adolescent agrees the pressure to marry has decreased due to the pandemic" are only presented for those unmarried adolescents who answered each of these questions at both rounds of the survey (n = 2,358 and n = 2,347 after excluding refusals, respectively).

populations. Still, this level suggests that a large number of adolescents are suffering, and this quantitative finding is triangulated in the qualitative data. A 17-year-old Syrian refugee girl from a host community noted:

> *"I am now reading a book that discusses symptoms of depression; I noticed that I have a number of symptoms. . . I sleep all day, I don't like talking to anyone, lack of laughter, I hate myself and. . . I do not like to seek help from anyone. . . I did not talk to the people closest to me, how will I talk to strangers?"*

Overall, the PHQ-8 improved by 3.2pp (p<0.001) between COVID-R1 and COVID-R2 for the panel sample of adolescents surveyed both times (Table 5, Panel B). Compared to adolescents living in host communities, Syrian adolescents living in ITSs and in camps showed significantly higher improvements in PHQ-8 scores as the pandemic progressed (Fig 1), likely reflecting greater access to services in camps and their tight-knit social fabric, as residents are predominantly from the same extended family and clan networks (discussed further, below). In the quantitative findings, in-school adolescents show larger declines in symptoms of depression compared to out-of-school adolescents (-5.4pp, p<0.05) with older adolescents also showing significantly larger improvements when compared with the younger cohort. The

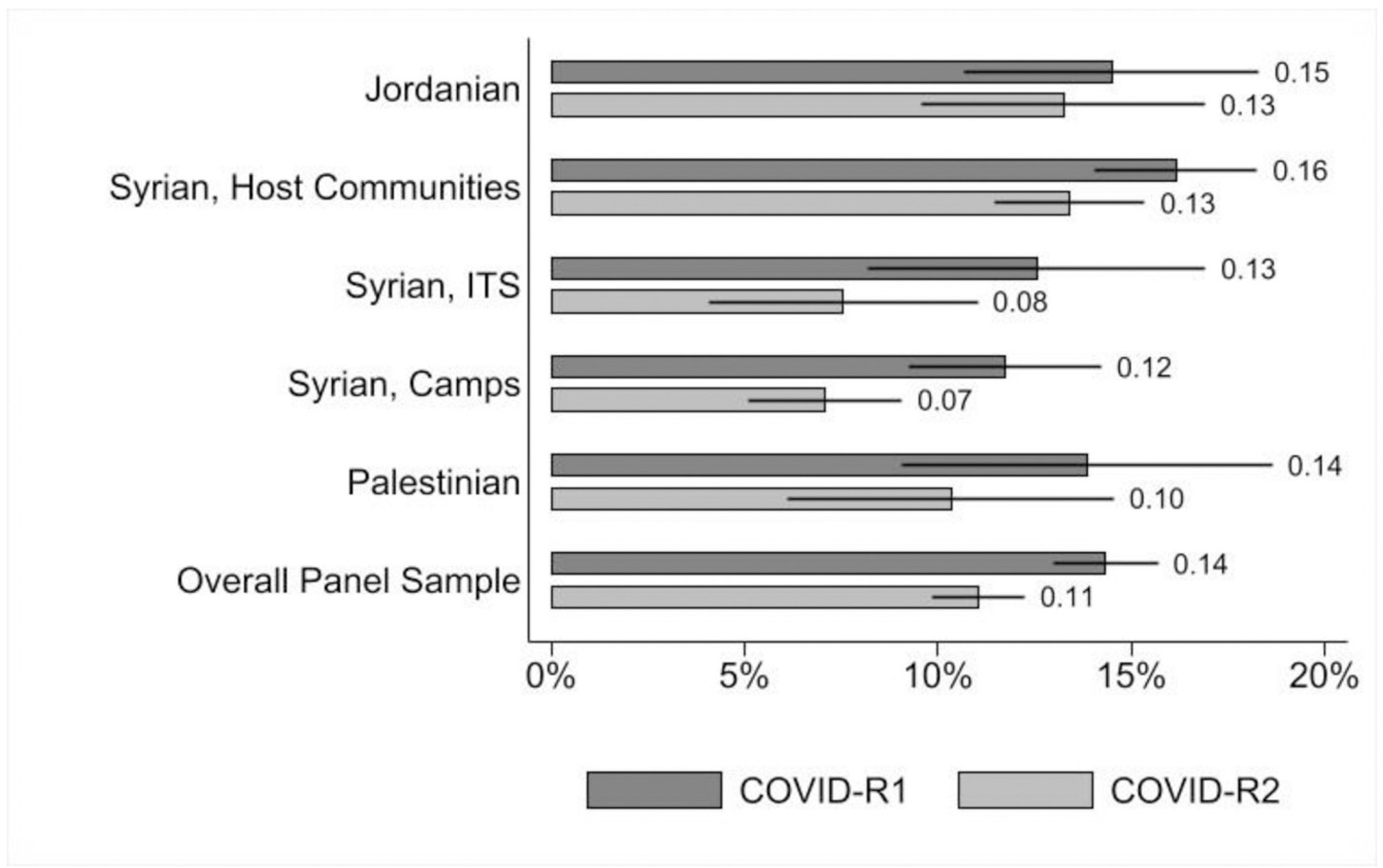

**Fig 1. Proportion of adolescents showing symptoms of moderate or severe depression, by nationality and location of residence (COVID-R1 and COVID-R2).**

qualitative findings suggest that this trend is partly because over time, the rigidity of the lock-down rules relaxed somewhat, and also young people became more accustomed to online schooling and interactions with peers. For older out-of-school adolescent boys, as the restrictions on mobility eased, they also were able to access more daily work opportunities and regain a sense of purpose. As a 15-year-old Syrian boy in a host community emphasized:

> "*I was lost when I couldn't go out and find construction work during the first lockdown to support my mother and my younger siblings. Now, times are still very hard, but I am doing something.*"

Perceptions of the broader community-level impacts of COVID-19 support the individual-level data. Among older adolescents, 83.1% reported that, as a result of COVID-19, the economic downturn, and service disruptions, people in their community "*were becoming very anxious or depressed*," and more than a quarter (27.9%) reported that, in their community, "*there is an increase in thoughts of self-harm, or people harming themselves*" compared to before the pandemic. Notably, the rate of adolescents identifying increased self-harm in their community was significantly higher among Jordanians than their Syrian counterparts. For example, Syrians living in camps were 7.7pp ($p<0.05$) less likely to report increased anxiety in the community, and 20.9pp ($p<0.001$) less likely to report increased thoughts of self-harm.

Echoing these quantitative findings—and high levels of stigma around psychosocial and mental health challenges in Jordanian culture notwithstanding—several adolescents in the qualitative data reported a heightened risk of suicide among young people in the community. An 18-year-old Syrian girl in a host community noted that: "*There are more suicides. . . For example, a youth in the neighbourhood was upset because of his living conditions and corona [virus], so he took a rope and hanged himself.*" This open discussion of mental ill-health is salient; several key informants noted that, in the Jordanian context, people tend to internalize psycho-emotional challenges and only vocalize them when they face extreme stress. This tendency is at least partly because, according to Islamic religious norms, suicide is considered *haram* (a major sin), and the families of people who commit suicide tend to be highly stigmatized in the community.

According to the GAD-7 scale, 12.4% of adolescents in the sample are experiencing symptoms of moderate-to-severe anxiety, with rates higher among older adolescents (4.5pp, $p<0.01$) and females (4.5pp, $p<0.001$) relative to their counterparts. The qualitative data underscored that anxiety was also linked to strong feelings of social isolation and, in some communities, fraying social cohesion, as one 17-year-old Syrian adolescent girl living in a refugee camp stated:

> "*When you talk to a person you feel that they are nervous a lot of the time. The person is annoyed and suffocated. Before corona[virus] people used to go out, but now there is no going out. There is nothing.*"

Qualitative data suggested that the finding on higher levels of anxiety among adolescent girls compared to adolescent boys is at least partly a result of less privacy, particularly in relation to menstrual hygiene (around which there are strong cultural taboos), as male family members were more often at home during lockdowns. These challenges to cultural norms were compounded by limited economic resources, inadequate water, and a lack of understanding among male family members about girls' needs, in terms of both privacy a well as securing supplies, given that their mobility is more restricted than that of boys and men. As a 16-year-

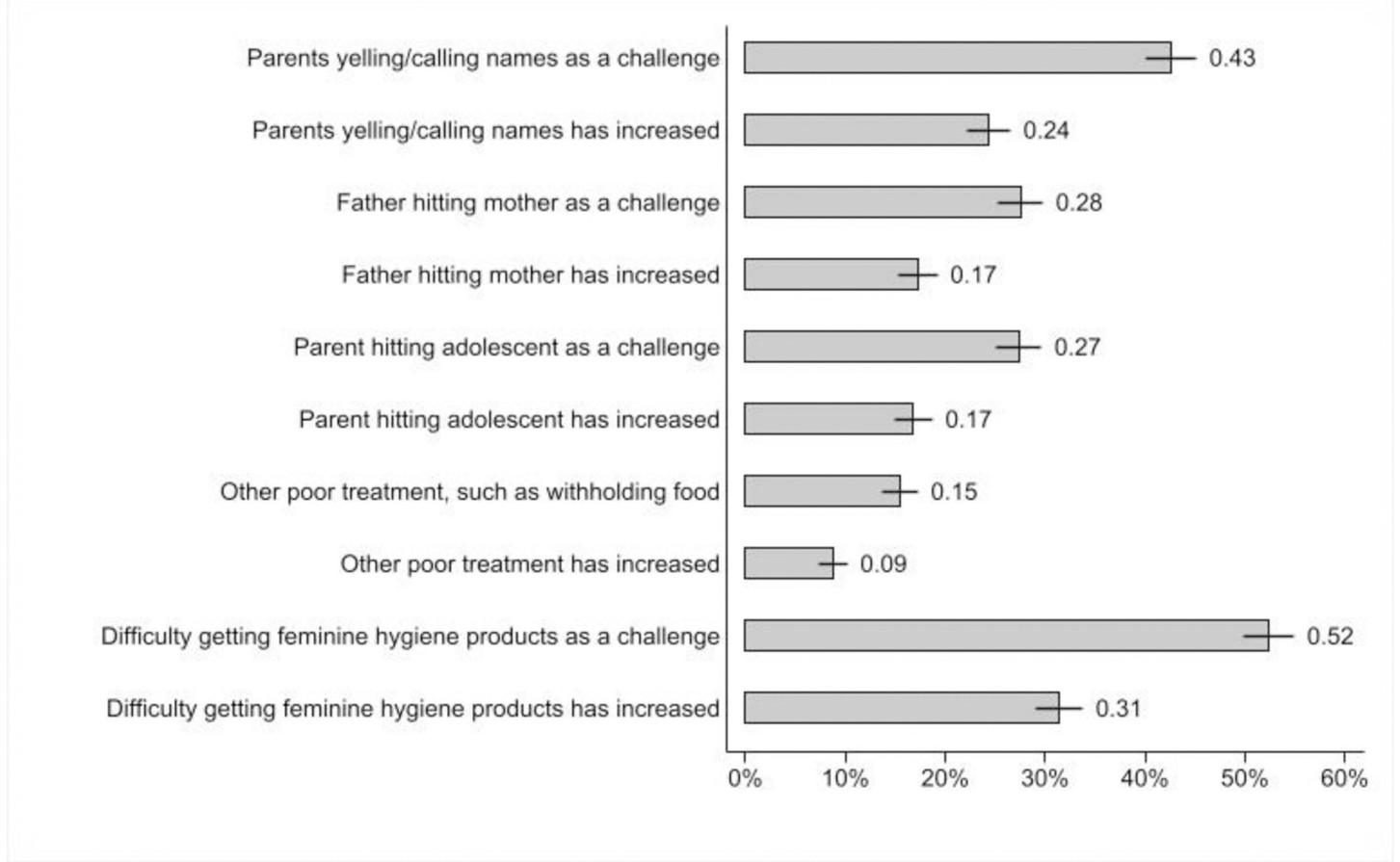

**Fig 2. Challenges around bodily integrity and increases in those challenges due to COVID-19 (unmarried adolescent girls).**

old Jordanian girl in a host community emphasized: "*For sure, it is a shame to tell [that I need menstrual supplies] to my brother or my father. We can't talk about private girls' issues with fathers, mothers or brothers.*" These findings are also borne out in the survey data. Fig 2 highlights that 52% of never-married females see accessing menstrual hygiene products as a challenge, with 60% of these adolescents noting that this challenge had increased during the pandemic.

The surveys also explored whether adolescent girls felt increased pressure to marry as a result of the pandemic. Overall, 16.5% of adolescents (both boys and girls) reported that they worried about marrying earlier because of the pandemic, with no significant difference between boys and girls after controlling for other factors (Table 5, Panel A). However, in the panel sample, the percentage of adolescents who worried about marrying earlier increased by 2.0 points across the two survey rounds, particularly for younger adolescents and adolescents from wealthier households (Table 5, Panel B). That said, a larger share of adolescents (if still a minority) reported that they were under less pressure to marry as a result of the pandemic (24.8%), with the panel sample displaying no change between rounds. This finding was less common for girls, who were 6.6 percentage points less likely than boys to report decreased pressure to marry (p<0.001) after controlling for other factors.

Qualitative data suggested that this decrease in pressure to marry was not due to shifting gender norms around the desirability of child marriage, but more a result of deepening

economic vulnerabilities, which make it more difficult to cover the costs involved in holding a wedding and setting up a new household. An 18-year-old Syrian refugee adolescent boy living in a host community encapsulated the challenge as follows:

> "*I can't save money to marry and I can't save money to be able to rent a house when I get married. During these conditions, it is difficult to get engaged, to pay money for dowry, to get married, and to buy the furniture for the house also.*"

The qualitative findings also indicated that reduced pressure to marry early is partly due to social distancing requirements, which prevent people organizing large wedding celebrations. It seems some girls and mothers are also reluctant to forego such an important social event and the status it confers among extended family and community. As a 16-year-old Syrian refugee girl in a host community explained, "*I don't want a corona wedding.*"

## Challenges to coping and resilience under COVID-19

The remaining findings focus on the positive and negative coping mechanisms employed by adolescents during the pandemic. The mixed-methods data suggested that interconnected factors at the individual, family, and community levels, along with policy and programming responses, shape how the pandemic may be influencing adolescent psychosocial wellbeing.

**Individual-level coping repertoires and resilience.** Table 6, Panel A provides an overview of adolescent coping strategies, both overall and in response to the pandemic. Looking at findings from the Brief Resilient Coping Scale (BRCS) (only measured at COVID-R2), there is a clear dichotomy in our sample, with 35.2% of adolescents considered to have low resilience and 15.9% high resilience. Across most subgroups, prevalence of high and low resilience looks similar, apart from notably higher levels of high resilience (4.4pp, p<0.01) and lower levels of low resilience (-7.0pp, p<0.01) among adolescents enrolled in school before the pandemic. This is also the case among older adolescents with 3.9pp (p<0.05) higher high resilience and 12.8pp (p<0.001) lower low resilience. Married adolescents (8.9pp higher than non-married adolescents, p<0.01) and Syrians in camps (6.7pp higher than Jordanians, p<0.01) also have higher levels of high resilience.

Qualitative data underscores the creative strategies that many young people in our sample have employed to withstand the effects of the pandemic, including reading, diary-writing, games with siblings, and supporting parents with household chores. A number of Syrian refugees also reflected that because of the trauma and deprivation they had experienced at a young age due to forced displacement, poverty, and discrimination by the host population in Jordan, they had learnt to become adept at coping with repeated exposure to shocks and to adopt an intentional approach to resilience. As a 16-year-old Syrian refugee boy from a refugee camp reflected:

> "*At the moment, the psychological status among people has changed—it can be affected quickly by any little trigger. But it is a decision to change your psychological status. It is normal. I spend my time kidding around with my siblings, talking with my friends, talking to my father, to my brother and to my uncle.*"

Several adolescent girls also highlighted that creative approaches to expressing their emotions and frustrations, such as drawing and diary-writing, were key to a positive coping approach. As a 16-year-old Jordanian girl in a host community noted: "*I have started to write stories and diaries after corona[virus]. I show them to my family and my relatives.*"

**Table 6. Adolescent coping strategies.**

Panel A. Outcomes reported at COVID-R1 and/or COVID-R2 (among 3,311 adolescents interviewed at either round)

| Covariates | Low resilient coping (BRCS 0–9), COVID-R2 only | High resilient coping (BRCS 13–16), COVID-R2 only | Adolescent reports coping well with stress caused by pandemic, COVID-R2 only | Adolescent reports seeking comfort and guidance from religion, COVID-R2 only | Adolescent ever smoked cigarettes (males age 15+ only) | Increased smoking in pandemic (among regular smokers, males age 15+ only) | Decreased smoking in pandemic (among regular smokers, males 15+ only) |
|---|---|---|---|---|---|---|---|
| = 1 if household had above-median assets at baseline | -0.018 | 0.008 | 0.024 | -0.007 | -0.009 | -0.031 | -0.025 |
| | (0.019) | (0.015) | (0.019) | (0.012) | (0.036) | (0.062) | (0.067) |
| = 1 if household is Syrian and lives in host community | -0.045 | 0.029 | 0.035 | 0.048* | -0.311*** | 0.037 | -0.046 |
| | (0.029) | (0.021) | (0.029) | (0.019) | (0.065) | (0.092) | (0.099) |
| = 1 if household is Syrian and lives in ITS | 0.022 | 0.035 | -0.022 | 0.049 | -0.468*** | 0.114 | -0.195 |
| | (0.043) | (0.031) | (0.043) | (0.026) | (0.088) | (0.141) | (0.145) |
| = 1 if household is Syrian and lives in camp | -0.046 | 0.067** | 0.060* | 0.032 | -0.311*** | 0.015 | -0.064 |
| | (0.031) | (0.023) | (0.030) | (0.020) | (0.067) | (0.094) | (0.101) |
| = 1 if household is Palestinian | 0.049 | 0.011 | -0.053 | 0.030 | -0.075 | -0.044 | 0.083 |
| | (0.041) | (0.028) | (0.041) | (0.026) | (0.096) | (0.122) | (0.128) |
| = 1 if adolescent girl | 0.034 | -0.024 | 0.001 | 0.011 | – | – | – |
| | (0.018) | (0.014) | (0.018) | (0.011) | | | |
| = 1 if adolescent was enrolled in formal school immediately before pandemic | -0.070** | 0.044* | 0.040 | 0.011 | -0.209*** | -0.086 | 0.083 |
| | (0.025) | (0.020) | (0.026) | (0.016) | (0.035) | (0.061) | (0.066) |
| = 1 if adolescent was ever married (at any survey round) | -0.014 | 0.089** | 0.046 | 0.039 | 0.024 | -0.053 | 0.215 |
| | (0.038) | (0.033) | (0.038) | (0.020) | (0.109) | (0.167) | (0.168) |
| = 1 if adolescent is in younger cohort (10–14 years) | 0.128*** | -0.039* | -0.053** | 0.006 | – | – | – |
| | (0.020) | (0.016) | (0.020) | (0.012) | | | |
| Sample mean (sd) | 0.352 | 0.159 | 0.655 | 0.907 | 0.378 | 0.367 | 0.502 |
| | (0.478) | (0.366) | (0.475) | (0.290) | (0.485) | (0.483) | (0.501) |
| Observations | 2885 | 2885 | 2885 | 2886 | 791 | 289 | 289 |

Panel B. Change in outcomes reported in COVID-R1 v. COVID-R2 (among 2,574 adolescents interviewed both rounds)

| Covariates | Change in adolescent self-reports ever smoking cigarettes (males, age 15+ only) | Change in adolescent reports increased smoking in pandemic (regular smokers among males age 15+ only) | Change in adolescent reports decreased smoking in pandemic (regular smokers among males age 15+ only) |
|---|---|---|---|
| = 1 if household had above-median assets at baseline | 0.041 | -0.030 | 0.037 |
| | (0.039) | (0.138) | (0.119) |
| = 1 if household is Syrian and lives in host community | 0.027 | -0.210 | -0.023 |
| | (0.066) | (0.168) | (0.150) |
| = 1 if household is Syrian and lives in ITS | -0.003 | 0.179 | -0.173 |
| | (0.083) | (0.238) | (0.226) |

*(Continued)*

|  |  |  |  |
|---|---|---|---|
| = 1 if household is Syrian and lives in camp | 0.056 | -0.226 | -0.250 |
|  | (0.069) | (0.178) | (0.156) |
| = 1 if household is Palestinian | 0.162 | -0.068 | 0.209 |
|  | (0.101) | (0.300) | (0.249) |
| = 1 if adolescent was enrolled in formal school immediately before pandemic | -0.035 | 0.045 | -0.197 |
|  | (0.037) | (0.130) | (0.117) |
| = 1 if adolescent was ever married (at any survey round) | 0.128 | 0.268 | -0.225 |
|  | (0.093) | (0.238) | (0.237) |
| Panel mean for outcome at COVID-R2 (sd) | 0.338 | 0.356 | 0.356 |
|  | (0.473) | (0.480) | (0.479) |
| Panel mean change between rounds (sd) | 0.062*** | 0.208*** | -0.256*** |
|  | (0.380) | (0.626) | (0.552) |
| Observations | 535 | 125 | 125 |

ds at COVID-R1 and/or COVID-R2 regressed on the listed covariates. For each outcome, adolescents who refused a question or responded "Don't know" are excluded from the sample for that outcome. Note that the outcomes "Low resilient coping score", "High resilient coping score", "Adolescent reports coping well with stress caused by the pandemic", and "Adolescent reports seeking comfort and guidance from religion" are drawn from the Brief Resilient Coping Scale (BRCS), administered at COVID-R2 only (n = 2,885 to n = 2,886 after excluding refusals). Questions about smoking cigarettes were only asked to adolescent males aged 15 and older; rates of ever smoking are among all males aged 15 and older who provided a response (n = 791 after excluding refusals), while rates of increased and decreased smoking are among males who self-reported being regular smokers in the survey (n = 289 after excluding refusals).

Notes: Standard errors of covariates in parentheses; * p<0.05

** p<0.01

*** p<0.001. This panel presents the results of a linear multivariate regression of the change in key outcomes reported by adolescents and households interviewed at both COVID-R1 and COVID-R2 on the listed covariates. For each outcome, adolescents who refused a question or responded "Don't know" are excluded from the sample for that outcome. Note that the outcomes around smoking cigarettes were only asked to adolescent males aged 15 and older; changes in if adolescents report ever smoking cigarettes are reported among all males aged 15 and older who provided a response at both COVID-R1 and COVID-R2 (n = 535 after excluding refusals), while rates of increased and decreased smoking are among males who self-reported being regular smokers and provided responses about changing patterns in smoking at both COVID-R1 and COVID-R2 (n = 125 after excluding refusals).

Our mixed-methods data highlighted that most young people are coping by seeking comfort and guidance from religion (90.7% in the quantitative survey). Our qualitative research echoed these findings: as a 14-year-old Syrian boy from a refugee camp explained: "*Me and my friends go to pray in the mosque and ask for blessings from God to help us get over this... People started becoming more religious after corona[virus]."* While adolescent girls reported that they are largely limited to prayer at home due to gendered religious norms, adolescent boys noted that after the initial strict lockdown in March and April 2020, they had been able to attend the mosque and interact with neighbours and peers when they did so.

Another coping response with important health implications is tobacco and substance use. Over one-third (37.8%) of older boys reported ever having smoked cigarettes. Among smokers, 36.7% reported increasing the amount they smoked during the pandemic, with rates increasing across the two rounds among those in the panel sample (20.8pp, p<0.001). The qualitative data suggested that this pattern of behaviour may in part have been driven by stress. A 19-year-old Syrian boy from a host community noted that:

"*Smoking has increased among many peers. Where someone was smoking one pack, now he smokes two packs. The reason is that smoking could get the stress out or, if the person feels upset, he smokes more to get feelings of frustration out during the pandemic.*"

Importantly, however, a larger percentage (50.2%) of older boys reported smoking fewer cigarettes or quitting entirely, likely due to economic constraints and declining informal sector work opportunities for adolescents (but this rate declined substantially for the panel sample between COVID-R1 and COVID-R2: -25.6pp, p<0.001). Our mixed-methods data provided limited evidence for any increase in use of marijuana or harder drugs, although perceived community use (pre- and post-COVID-19) was relatively high: 16.7% of adolescents aged 15 and older interviewed at COVID-R2 agreed that peers of their own gender smoke marijuana in the community, and 9.7% agreed that peers use illicit drugs such as heroin or synthetic hashish/marijuana known as "joker".

**Family- and community-level coping and resilience: Social connectedness.** We now discuss the extent to which adolescent social connections with family and community networks are supportive of their coping and resilience during the pandemic.

*Family relationships.* Most in-school adolescents (formal school) reported receiving some form of support from their family to continue learning during school closures at either COVID-R1 or COVID-R2 (91.4%), including: support with schoolwork (67.5% among those with any family support); a space to study (79.1%); and/or a device with internet access (79.3%). Almost half (65.5% among those with any family support) acknowledged that their family was reducing their chore load so that they could keep up with their studies. The qualitative data highlights the important role that family relationships have played for some adolescents in moderating their psychosocial wellbeing and mental health during the pandemic, and in strengthening communication between parents and adolescents. As one 16-year-old Syrian refugee living in a host community reflected:

"*During this time I spent a lot of time at home. I got to know my family more than before... My father and I became friends now. Before the quarantine I did not talk to my father much. I always ran away from him—I was afraid of him. But now he is discussing and talking with me.*"

Key informants further noted that such family support is especially important, given the weak formal psychosocial support services available.

On the other hand, quantitative findings show that, for two-thirds of all adolescents in the sample (66.8%), household stress had increased during the pandemic, especially for Syrian adolescents in host communities (at least 10.7pp higher than any other group, p<0.001) and for adolescents who were in school prior to the pandemic (7.9pp, p<0.001) (Table 7, Panel A). When asked about stressors for other adolescents of the same age and gender in their community, respondents often identified a lack of resources (such as insufficient money to buy non-food items) as a challenge that had worsened during the pandemic (62.8%). The qualitative findings revealed that a number of young people, predominantly adolescent boys, were seeking exit options from Jordan—including migration through brokers—as a result of the extreme vulnerability they were facing. As a 14-year-old Syrian refugee boy in a host community emphasized: '*A lot of people are thinking of immigration. We want to immigrate [out of Jordan]. . . We want to immigrate in order to live.*"

More broadly, to understand changes in intra-household violence as well as access to services, we designed a vignette that asked a series of questions around what kinds of challenges "adolescents like you in the community are experiencing", and whether those challenges had increased, decreased or stayed the same since the onset of the pandemic. Fig 2 highlights the verbal, physical, and emotional violence that unmarried girls typically experience in these communities. It also shows that many adolescents perceived that violence had increased since the onset of the pandemic, with over 50% of adolescents who had experienced a given type of violence indicating that it had increased during the pandemic. The qualitative findings similarly underscore that elevated levels of household stress—stemming from economic pressures, unemployment, and fathers and brothers being significantly more home-bound than in pre-pandemic times—are driving increased rates of intra-household violence. As a 14-year-old Jordanian girl in a host community emphasized: "*Because of COVID everyone is sad and upset and nervous. Before, I didn't fight with my brothers and sisters. Now, we always fight.*"

*Peer relationships.* Looking at peer relationships, our quantitative survey showed that 57.3% of adolescents reported having a friend they could trust. However, girls—and particularly younger girls and married girls—were less likely to report having peer support, probably reflecting girls' more restricted mobility and higher levels of social isolation due to pre-existing discriminatory gender norms. Of all adolescents in the sample, 58.0% had interacted with friends or someone outside the household in the past week in-person at either COVID-R1 or COVID-R2, with significantly lower rates for Syrians. Girls also were 11.6pp (p<0.001) less likely to see non-household members in person than boys. Within the panel sample, the rate of adolescents interacting with others outside of the household had increased between survey waves (14.4pp, p<0.001), but remained at only 48.0% as of the COVID-R2 survey. Among adolescents surveyed at COVID-R2, almost a third (29.4%) had had no interaction with friends either in-person or online in the past seven days, which is very concerning, given the importance of peer interactions during the adolescent years. The rates of social isolation were higher among girls than boys (13.4pp, p<0.001) and married adolescents—who are mainly girls—in particular (7.5pp, p<0.05), with higher rates among poorer households (7.0pp, p<0.001).

Qualitative data underscored the emotional deprivation that adolescents have experienced as a result of being cut off from work- or school-based peers. As a 16-year-old Syrian refugee boy living in a host community emphasized:

"*My friends were like my brothers; I miss them so much. We used to go play in the football round, entertain ourselves, laugh together. We are no longer able to see each other with the COVID-19 curfew. This has affected me. . . having no one to get my back. My school was like home. I could always see my friends. Now, I feel extremely alone. I feel there is no taste to life now.*"

**Table 7. Social connectedness.**

Panel A. Outcomes reported at COVID-R1 and/or COVID-R2 (among 3,311 adolescents interviewed at either round)

| Covariates | Adolescent had family support for schooling (among enrolled pre-pandemic) | Household had contact with a teacher in past week (among enrolled pre-pandemic) | Adolescent reports that household stress increased | Adolescent has friend they can trust (COVID-R2 only) | Adolescent interacted with friends or non-household family in-person in past week | Adolescent had no contact with friends in past week, in-person or virtually (COVID-R2 only) | Adolescent has a personal mobile device with internet access | Adolescent interacted with friends virtually in past week (COVID-R2 only) | Adolescent self-reports increased technology access | Household currently receives any cash or food aid | Household cash or food aid decreased since pandemic began (among households receiving aid) |
|---|---|---|---|---|---|---|---|---|---|---|---|
| = 1 if household had above-median assets at baseline | 0.019 (0.012) | 0.070** (0.022) | -0.001 (0.018) | 0.053** (0.020) | 0.011 (0.019) | -0.070*** (0.018) | 0.106*** (0.017) | 0.069*** (0.020) | 0.061*** (0.016) | -0.024** (0.009) | -0.006 (0.020) |
| = 1 if household is Syrian and lives in host community | -0.023 (0.015) | -0.011 (0.029) | 0.107*** (0.026) | -0.019 (0.030) | -0.118*** (0.026) | 0.035 (0.027) | -0.075** (0.026) | -0.009 (0.030) | -0.002 (0.023) | 0.889*** (0.013) | 0.046 (0.086) |
| = 1 if household is Syrian and lives in ITS | -0.156*** (0.040) | -0.188*** (0.052) | -0.044 (0.040) | 0.044 (0.043) | -0.121** (0.040) | 0.038 (0.040) | -0.107** (0.037) | -0.033 (0.043) | -0.040 (0.037) | 0.889*** (0.017) | -0.078 (0.091) |
| = 1 if household is Syrian and lives in camp | -0.015 (0.016) | -0.014 (0.031) | -0.022 (0.028) | -0.016 (0.031) | -0.071* (0.028) | 0.019 (0.028) | -0.132*** (0.027) | -0.079* (0.031) | -0.036 (0.025) | 0.912*** (0.013) | -0.053 (0.086) |
| = 1 if household is Palestinian | -0.029 (0.023) | 0.020 (0.040) | -0.019 (0.037) | -0.046 (0.041) | 0.073* (0.035) | -0.011 (0.036) | -0.063 (0.035) | -0.069 (0.042) | -0.012 (0.033) | 0.277*** (0.032) | -0.384*** (0.096) |
| = 1 if adolescent girl | 0.028* (0.011) | 0.114*** (0.020) | -0.024 (0.017) | -0.049** (0.019) | -0.116*** (0.018) | 0.134*** (0.017) | -0.163*** (0.016) | 0.009 (0.019) | -0.035* (0.016) | 0.010 (0.008) | -0.018 (0.020) |
| = 1 if adolescent was enrolled in formal school immediately before pandemic | – | – | 0.079*** (0.023) | 0.043 (0.026) | 0.022 (0.024) | -0.042 (0.023) | -0.050* (0.022) | 0.036 (0.026) | 0.098*** (0.021) | 0.000 (0.010) | 0.031 (0.026) |
| = 1 if adolescent was ever married (at any survey round) | -0.189* (0.078) | -0.018 (0.082) | 0.013 (0.034) | -0.142*** (0.040) | 0.102** (0.036) | 0.075* (0.038) | 0.056 (0.035) | -0.085* (0.040) | -0.028 (0.034) | -0.062*** (0.018) | -0.049 (0.041) |
| = 1 if adolescent is in younger cohort (10–14 years) | 0.050*** (0.013) | 0.011 (0.021) | -0.055** (0.019) | -0.168*** (0.021) | 0.009 (0.020) | 0.116*** (0.019) | -0.361*** (0.019) | -0.198*** (0.021) | -0.110*** (0.017) | -0.005 (0.009) | -0.023 (0.022) |

(Continued)

**Table 7.** (Continued)

| | | | | | | | | | | |
|---|---|---|---|---|---|---|---|---|---|---|
| Sample mean (sd) | 0.914 | 0.576 | 0.668 | 0.573 | 0.580 | 0.294 | 0.432 | 0.560 | 0.737 | 0.787 | 0.623 |
| | (0.280) | (0.494) | (0.471) | (0.495) | (0.494) | (0.456) | (0.495) | (0.497) | (0.440) | (0.409) | (0.485) |
| Observations | 2420 | 2420 | 3310 | 2885 | 3310 | 2886 | 3311 | 2886 | 3310 | 3244 | 2553 |

**Panel B: Change in outcomes reported in COVID-R1 vs. COVID-R2 (among 2,574 adolescents interviewed both rounds)**

| Covariates | Change in family support for school closures during closures (among those enrolled) | Change in contact with teacher in past week (among those enrolled) | Change in adolescent reports that household stress increased | Change in adolescent interacted with friends or non-household family in-person in past week | Change in adolescent has a personal mobile device with internet access | Change in adolescent self-reported increased technology access | Change in household receipt of any cash or food aid | Change in aid decreased since pandemic began (among households receiving any aid) |
|---|---|---|---|---|---|---|---|---|
| = 1 if household had above-median assets at baseline | 0.005 | -0.048 | -0.013 | 0.002 | 0.038 | 0.025 | -0.005 | -0.011 |
| | (0.022) | (0.033) | (0.027) | (0.027) | (0.020) | (0.028) | (0.011) | (0.031) |
| = 1 if household is Syrian and lives in host community | -0.015 | 0.009 | -0.059 | 0.090* | -0.012 | 0.017 | 0.033 | 0.069** |
| | (0.031) | (0.047) | (0.040) | (0.043) | (0.032) | (0.042) | (0.018) | (0.025) |
| = 1 if household is Syrian and lives in ITS | 0.071 | 0.116 | -0.150** | 0.176** | 0.002 | 0.123* | 0.030 | 0.043 |
| | (0.066) | (0.080) | (0.057) | (0.059) | (0.046) | (0.060) | (0.023) | (0.049) |
| = 1 if household is Syrian and lives in camp | -0.042 | -0.067 | -0.041 | 0.181*** | 0.047 | 0.024 | 0.051* | 0.023 |
| | (0.032) | (0.049) | (0.043) | (0.044) | (0.033) | (0.043) | (0.021) | (0.030) |
| = 1 if household is Palestinian | -0.008 | 0.206** | -0.036 | 0.040 | 0.003 | 0.090 | 0.184*** | -0.024 |
| | (0.047) | (0.064) | (0.056) | (0.058) | (0.043) | (0.059) | (0.033) | (0.087) |
| = 1 if adolescent girl | 0.024 | -0.004 | 0.025 | -0.049 | -0.016 | -0.019 | 0.011 | -0.000 |
| | (0.021) | (0.030) | (0.026) | (0.026) | (0.019) | (0.026) | (0.010) | (0.030) |
| = 1 if adolescent was enrolled in formal school immediately before pandemic | – | – | 0.029 | 0.059 | -0.067** | 0.040 | 0.016 | 0.062 |
| | – | – | (0.036) | (0.036) | (0.029) | (0.036) | (0.015) | (0.040) |
| = 1 if adolescent was ever married (at any survey round) | -0.225 | -0.139 | -0.045 | 0.078 | 0.048 | 0.021 | 0.018 | -0.003 |
| | (0.138) | (0.141) | (0.054) | (0.059) | (0.043) | (0.055) | (0.036) | (0.074) |
| = 1 if adolescent is in younger cohort (10–14 years) | 0.019 | 0.062 | 0.058* | 0.044 | -0.073*** | 0.019 | -0.012 | 0.026 |
| | (0.023) | (0.033) | (0.029) | (0.029) | (0.022) | (0.029) | (0.011) | (0.033) |
| Panel mean for outcome at COVID-R2 (sd) | 0.832 | 0.361 | 0.497 | 0.480 | 0.361 | 0.541 | 0.814 | 0.497 |
| | (0.374) | (0.480) | (0.500) | (0.500) | (0.481) | (0.498) | (0.389) | (0.500) |
| Panel mean change between rounds (sd) | -0.034** | -0.126*** | -0.035** | 0.144*** | 0.072*** | -0.071*** | 0.022*** | 0.042** |
| | (0.440) | (0.639) | (0.628) | (0.627) | (0.467) | (0.639) | (0.249) | (0.615) |

(*Continued*)

**Table 7.** (Continued)

| Observations | 1768 | 1765 | 2568 | 2571 | 2572 | 2566 | 2335 | 1803 |
|---|---|---|---|---|---|---|---|---|

Notes: Standard errors of covariates in parentheses

* p<0.05

** p<0.01

*** p<0.001. This table presents the results of a linear multivariate regression of the outcomes of interest reported by interviewed adolescents and households at COVID-R1 and/or COVID-R2 regressed on the listed covariates. For each outcome, adolescents (or their adult female caregivers) who refused a question or responded "Don't know" are excluded from the sample for that outcome. Sample sizes for outcomes asked of the entire sample ("Adolescent reports that household stress has increased"; "Adolescent interacted with friends or non-household family in-person in the past week"; "Adolescent has a personal mobile device with internet access"; and "Adolescent self-reports increased technology access") range from n = 3,310 to n = 3,311 after excluding refusals. Note that the outcomes on family support for schooling and household contact with a teacher in the past week are among students who were enrolled in formal school only (n = 2,420). The following outcomes were only collected at COVID-R2 (n = 2,885 to n = 2,886 after excluding refusals): "Adolescent has a friend they can trust"; "Adolescent had no contact with friends in the past week", "Adolescent interacted virtually with friends in the past week". Responses on receipt of cash or food aid are drawn from the adult female survey, which is available for 98% of adolescents who were interviewed at either COVID-R1 or COVID-R2 (adult female survey n = 3,244 after excluding refusals). Specifically, cash and food aid included in the survey include four major programmes: World Food Programme (WFP) food vouchers; United Nations Relief and Works Agency for Palestine Refugees in the Near East (UNRWA) food aid cards; UNHCR cash transfers; and Hajati cash transfers. On the final outcome "Household cash or food aid decreased since pandemic began", the table presents results only for those households that reported receiving any aid from the four programmes (n = 2,553 after excluding refusals).

Notes: Standard errors of covariates in parentheses

* p<0.05

** p<0.01

*** p<0.001. This panel presents the results of a linear multivariate regression of the change in key outcomes reported by adolescents and households interviewed at both COVID-R1 and COVID-R2 on the listed covariates. For each outcome, adolescents who refused a question or responded "Don't know" are excluded from the sample for that outcome. For questions presented to the entire sample, the sample size ranges from n = 2,566 to n = 2,572 after excluding refusals. The change in reports of family support for schooling and household contact with a teacher in the past week are among students who were enrolled in school who responded to these questions at both rounds (n = 1,768 and n = 1,765, respectively after excluding refusals). Responses on receipt of cash or food aid are drawn from the adult female survey, which is available at both COVID-R1 and COVID-R2 for 92% of adolescents who were also interviewed at both COVID-R1 and COVID-R2 (adult female survey n = 2,361 in this panel, n = 2,335 for this outcome after excluding refusals). On the final outcome "Change in household cash or food aid decreased since pandemic began", the table presents results only for those households in the panel that reported receiving any aid from the four programmes (n = 1,803).

Girls, in particular, highlighted that, with formal and informal educational services closed, they lacked a culturally acceptable safe space in which to interact with peers, and that interactions with neighbours (which boys might still be able to have) were not an option. A 15-year-old Palestinian refugee girl living in a camp noted:

"*Nowadays we can't go to school to meet our friends; we can't go to the centres and we don't have any activities. We just do our schoolwork on our phone. Before, we used to participate in the centres—it was a space for us to be normal and free. Nowadays, this is forbidden.*"

*Online networks.* Online peer networks emerged as an important part of life for adolescents during the pandemic, especially in host communities, where the internet is more widely accessible. More than two-thirds (73.7%) of the sample reported that they had increased access to technology compared to before the pandemic, with rates increasing over time. But rates were lower for girls and adolescents who were not enrolled in school. Moreover, while 42.2% of adolescents had their own personal device with internet access, these rates were substantially lower among Syrian refugees compared to Jordanians across host communities (-7.5pp, p<0.01), ITSs (-10.7pp, p<0.01) and camps (-13.2pp, p<0.001), among girls compared to boys (-16.3pp, p<0.001), and among households with below-median assets compared to those with above-median assets (-10.6pp, p<0.001).

Over half of adolescents (56.0%) had interacted with friends virtually in the past week, including messaging by phone, connecting over social media, or playing online games. The qualitative data suggested that, for many young people, access to online peer networks had been a critical part of their coping repertoire during the pandemic. As a key informant noted:

"*During this tense period, the adolescents used to spend their time on smartphones only. . . I mean, they spend their time on smartphones such as playing games, watching YouTube, and playing new games that they play together as a group. . . This is their coping strategy.*"

That said, of all online users (85% of the sample interviewed at COVID-R2), 11.9% reported having had a negative or uncomfortable online experience in the past 12 months, with higher rates among boys than girls (15.2% vs. 8.8%). This was also echoed in the qualitative data. A 17-year-old Jordanian boy from a host community explained his experienced as follows:

"*Disputes and insults occur among students on WhatsApp groups. It is usually over small and simple things but I stopped interacting on the WhatsApp group because of this—and so I dropped out of the whole study group. It was stressful.*"

*Relationships with teachers.* Support from teachers during remote learning was reported to be limited. Only 57.6% of in-school adolescents interviewed at either round reported having had any contact with a teacher in the seven days prior to the survey; among those surveyed at COVID-R2, only 19.7% reported receiving feedback from a teacher in the past week. Girls and adolescents from wealthier households were more likely to have interacted with a teacher than boys. Qualitative data suggested that some girls enjoyed strong communication with their teachers before the pandemic, and that the teachers had continued to provide emotional support during school closures. A 17-year-old Syrian refugee girl, for example, noted:

"*My teacher at school gives us a course online. . . to help us know who we are from inside and what is bothering us. . . I communicate with all my teachers during this period. I have very good relationships with my teachers.*"

However, others noted that the level of emotional support they had received at school was no longer possible during the pandemic, and that all-online class groups did not provide sufficient opportunities for more personal interactions with teachers. As a 12-year-old Syrian girl living in a refugee camp stated:

"*I used to tell the teacher anything wrong that happened to me. . . and she would advise me. . . But I haven't been able to talk to her by phone at all. . . They used to send us homework to solve on WhatsApp but there are so many girls in the group, so now you cannot say anything because everyone would then know. So I don't say anything.*"

*Policy and programmatic level*: *Social assistance and youth empowerment programming.* Alongside peer, family, and school support, social assistance and empowerment programming may also influence adolescent outcomes during COVID-19. Our quantitative findings showed that, while 78.7% of the sample lived in a household that regularly received aid from the Jordanian government or an international organization—including the World Food Programme (WFP), UNHCR, or the United Nations Relief and Works Agency for Palestine Refugees in the Near East (UNRWA)—over half (62.3%) of these households received less support during the pandemic. For many young people, the levels of social protection were inadequate to prevent hunger and food insecurity. More than a third (37.3%) of all adolescents reported having been hungry in the past four weeks at the time of the survey. Fig 3 highlights that rates of hunger increased for all nationalities at the onset of the pandemic, with limited signs of recovery (except for Jordanians and for Syrians living in camps).

Qualitative data confirmed this finding, with some adolescents reporting that—in the context of pre-existing poverty and hardship, and the absence of adequate emergency social assistance—they perceived a marked increase in the numbers of young people in host communities turning to risky coping approaches (such as begging) to secure basic resources for themselves and their families. As a 17-year-old Jordanian girl living in a host community observed:

"*Beggars come to us while we are shopping. Sometimes girls in groups of five or six, sometimes alone. They are about 18 or 19. . . Nowadays, many people have to beg, especially females. Young women whose husbands were working in a company before corona[virus] and now his work has stopped. . . She has to beg to spend on herself and her young children.*"

However, in line with broader findings from WFP in 2020 [90], our qualitative data also underscored that adolescents in Azraq camp were at heightened risk of food insecurity as families were unable to supplement their household income through informal work due to the lockdown, which was easier to enforce in Azraq given its location in a remote desert area. A 14-year-old Syrian refugee boy living in Azraq camp explained the situation as follows:

"*Corona[virus] made people face hunger. . . When the corona situation became serious, money started finishing and people started running away from the camp as they were dying of hunger. They closed the camp for two to three months and you couldn't go out to work.*"

Moreover, as this quote underscores, the reduced aid available to some families during the pandemic noted above was an important determinant of household vulnerability and may explain some of the transmitted mental health stresses reported.

Turning to adolescent-specific programming, quantitative regression analysis incorporating nearest neighbour matching confirmed the protective role of Makani centres—national

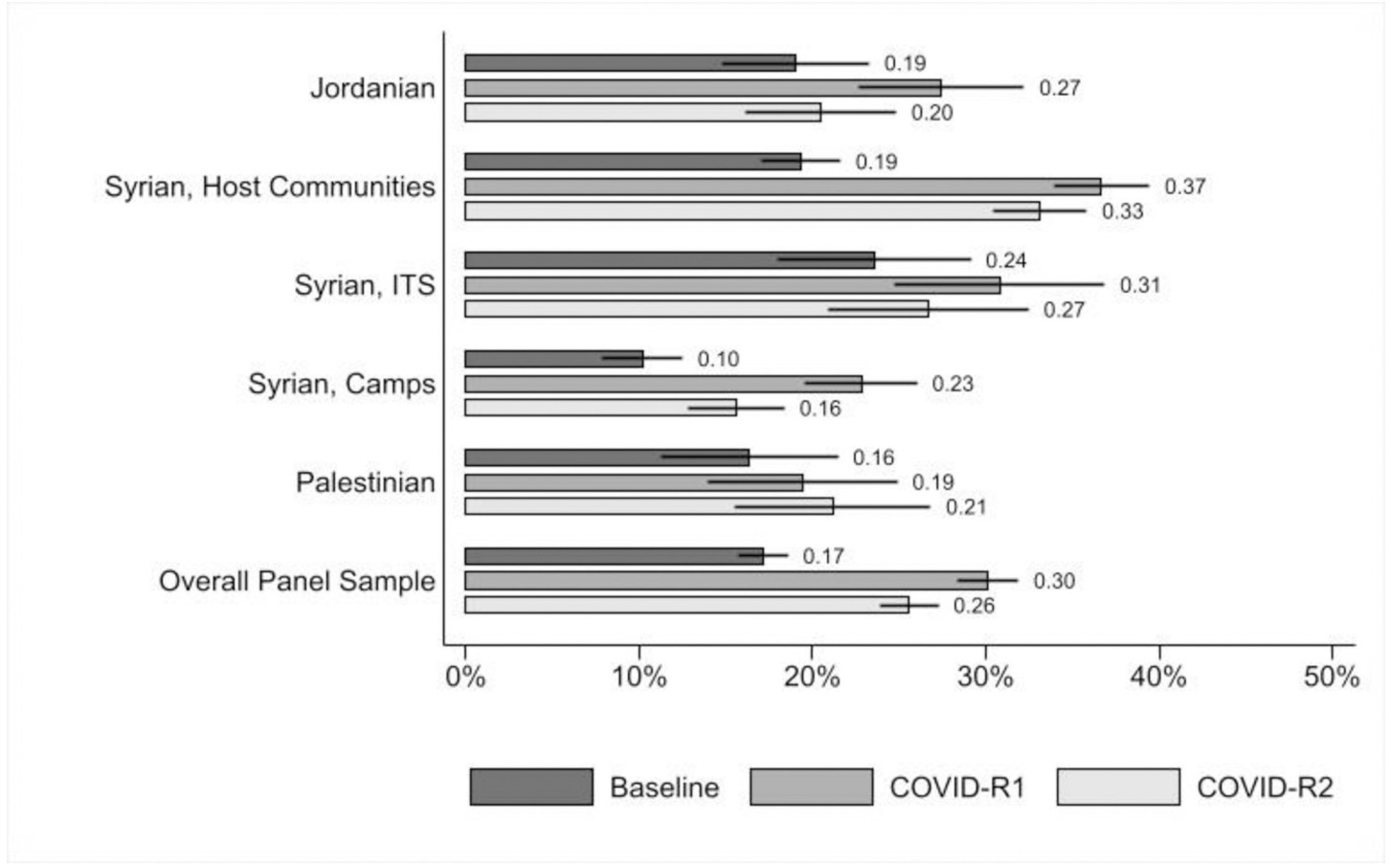

**Fig 3. Adolescents experiencing hunger in the past four weeks (baseline, COVID-R1, and COVID-R2), by nationality and location of residence.**

one-stop child and adolescent Makani centres managed by UNICEF and run through community organizations—on certain outcomes, with limited evidence of impact for others (Table 8).

While we did not see any impact of participation in Makani on reducing symptoms of depression, anxiety or overall coping, participation did appear to improve support networks and access to formal education. Makani participation was associated with a 5.2pp reduction (p<0.01) in the likelihood that the adolescent stayed at home for the past seven days, a 4.1pp (p<0.05) reduction in the likelihood of not interacting with friends in the past 7 days, and a 7.0pp (p<0.01) increase in the likelihood that the boy/girl had a trusted friend. The impacts on interacting with friends were larger for girls and for Syrians in host communities. In addition, for girls, Makani participation was associated with a 7.2pp (p<0.05) increase in reporting that adults outside the household were helping her cope with the pandemic and a 5.4pp (p<0.05) reduction in the likelihood of receiving less support during the pandemic.

Makani also promoted formal learning outcomes across all analysed subgroups, with adolescents who attended Makani 9.6pp (p<0.001) more likely to be enrolled in school when the pandemic started, and 9.3 (p<0.001) more likely to have returned to school when schools reopened briefly in September 2020.

Similarly, qualitative data suggests that the Makani centres have played an important supportive role during the pandemic for those who already were enrolled at the centres. Centre facilitators—who were able to rely on trusted relationships with adolescents and their

**Table 8. Average treatment effect (ATE) of Makani participation on outcomes of interest at COVID-R2 survey using nearest neighbor matching.**

| | Overall | Girls | Syrians (Host) | Syrians (Camp) |
|---|---|---|---|---|
| = 1 if Patient Health Questionare-8 (PHQ-8) Score > = 10 (suggest symptoms of moderate to severe depression) | -0.004 | -0.011 | 0.011 | -0.011 |
| | (0.013) | (0.020) | (0.021) | (0.022) |
| = 1 if Generalized Anxiety Disorder-7 (GAD-7) Score > = 10 (suggests symptoms of moderate to severe anxiety) | -0.004 | -0.009 | 0.009 | -0.007 |
| | (0.014) | (0.022) | (0.022) | (0.033) |
| = 1 if adolescent scored within low resilient coping range (0–9), Brief Resilient Coping Scale | -0.026 | -0.040 | -0.016 | -0.011 |
| | (0.021) | (0.028) | (0.027) | (0.047) |
| = 1 if adolescent's friends are helping him/her cope with the pandemic | 0.028 | 0.049 | -0.013 | 0.079 |
| | (0.021) | (0.031) | (0.029) | (0.049) |
| = 1 if other adults outside the family are helping him/her cope with the pandemic | 0.041 | 0.072 | 0.015 | 0.034 |
| | (0.022) | (0.031)* | (0.030) | (0.052) |
| = 1 if adolescent gets less support during COVID-19 | -0.018 | -0.054 | -0.008 | -0.005 |
| | (0.018) | (0.024)* | (0.025) | (0.043) |
| = 1 if adolescent reports he/she has a friend that he/she can trust | 0.070 | 0.088 | 0.092 | 0.075 |
| | (0.022)** | (0.030)** | (0.030)** | (0.053) |
| = 1 if adolescent stayed home for the past 7 days | -0.052 | -0.049 | -0.046 | -0.074 |
| | (0.020)** | (0.031) | (0.025) | (0.048) |
| = 1 if adolescent had no interaction with friends either in person or online in past 7 days | -0.041 | -0.058 | -0.027 | -0.061 |
| | (0.019)* | (0.027)* | (0.028) | (0.037) |
| = 1 if adolescent enrolled in formal school at onset of COVID-19 | 0.096 | 0.091 | 0.072 | 0.171 |
| | (0.017)*** | (0.021)*** | (0.024)** | (0.039)*** |
| = 1 if adolescent returned to school when schools briefly reopened in September 2020 | 0.093 | 0.082 | 0.080 | 0.149 |
| | (0.018)*** | (0.024)** | (0.025)** | (0.044) |
| Sample Size | 2800 | 1441 | 1284 | 670 |

01, *** p<0.001. Each cell is the coefficient from a separate regression and provides an estimate of the difference in the outcome between Makani and non-Makani participants. Estimates are derived from matching techniques that utilize the 'teffects' command in Stata, which computes robust Abadie-Imbens standard errors, to implement nearest neighbor matching (using five matches per observation) to match participants to non-participants on a set of baseline variables arguably exogenous to Makani participation. These variables include the following baseline characteristics: asset decile, gender, age cohort, disability status, nationality, household size, whether the household is female headed, and if the household receives any aid. We require exact matches for gender, age cohort, nationality and location of residence.

caregivers—quickly pivoted to provision of online support, through WhatsApp and mobile phone text messages. A 17-year-old adolescent girl in Azraq camp expressed positive feedback on Makani's role:

> "*Makani groups are very useful, because most of the people in the camp are staying home, not going out or coming back in. This programme has a big role in helping us know everything that is going on while staying home.*"

Alongside access to programmes, the Jordanian government made a concerted effort to engage young people in adolescent empowerment activities (volunteering) as a means to strengthen resilience during the pandemic. Whereas just 4.4% of our sample were involved in youth volunteer activities in the early stages of the pandemic, our survey findings from December 2020 showed that 10.5% reported volunteer engagement, with higher rates among boys

than girls. Respondents in our qualitative research emphasized that this gender gap was possibly due to discriminatory gender norms and mobility restrictions on girls, stemming from fears of sexual harassment and threats to family honour. As a 13-year-old Syrian girl living in an informal tented settlement explained: *"I could not participate with any programme or organization. I'm not allowed to volunteer or go there without my brother or father."*

## Discussion

This article, based on panel data collected before and after the onset of the COVID-19 pandemic, aimed to explore the pandemic's effects on the psychosocial wellbeing and resilience of adolescents affected by forced displacement. It also explored differences in the patterning of these impacts, based on age, gender, residential location, nationality, wealth, schooling, and marital status. Drawing on a capabilities framework grounded in the theoretical work of Sen [68] and Nussbaum [69], the article utilized mixed-methods panel data from the GAGE longitudinal study to assess the effects of the pandemic and related service disruptions on adolescents' individual coping repertoires and on their social connectedness with peers and family [91]. To our knowledge, this is only the fifth peer-reviewed article that exploits panel data to look at the impact of COVID-19 on adolescent psychosocial wellbeing and resilience in LMICs (two of which only look at adolescents aged 18 and 19) [16, 92–94]. This article is also the first (as far as we are aware) that incorporates mixed-methods data and uses multiple COVID-19 survey rounds to explore this topic.

In line with an emerging evidence base on the effects of the pandemic in LMICs [92, 93, 95], our findings underscored that the pandemic negatively impacted adolescent psychosocial wellbeing in multiple domains. While many young people reported having adopted a range of coping strategies, only a small minority expressed high resilience. Moreover, as also observed elsewhere [28, 96], the patterning of stressors reinforced existing social and economic inequalities [97], with girls—and especially married girls, as well as adolescents from the poorest households and those who are out of school—facing greater deprivations [17, 27, 98].

Our mixed-methods research also contributes to the limited research on pandemic experiences among refugee populations [99–102], and especially limited research on adolescents in humanitarian settings [27, 103]. It highlights the less visible disadvantages experienced by vulnerable young people in host communities compared to those in refugee camps and, to a lesser extent, those in ITSs—and underscores that these disadvantages frequently cut across nationality and citizenship status. Recent analyses of the economic climate and labour market dynamics in Jordan and the Middle East and North Africa (MENA) region have recognized the precarious conditions of refugees and vulnerable local households living in host communities, in the context of the Syrian refugee crisis [104]. Our findings go further to provide novel data on the complex disadvantages facing adolescents, which have distinct patterning between camps and host community settings, and highlight how the pandemic is exacerbating these inequalities.

Our mixed-methods analysis elicited factors that support resilience for some adolescents, including trusted relationships with family and peers (especially online networks) and, to a lesser extent, with teachers (and for girls more so than boys). Somewhat strikingly, survey findings revealed higher levels of social connectedness among young people in camps than those in host communities. This does, however, resonate with pre-pandemic findings on relatively high levels of social capital among Syrian refugees living in ITSs in Lebanon, and the positive effects this has on psychosocial resilience [105]. Moreover, the host community disadvantage is not a matter of nationality; rather, it appears to reflect a greater degree of social isolation and less support from peers. This greater level of resilience among adolescents in camp settings

appeared to stem from a range of factors: more interaction with neighbours and extended family who lived in closer proximity; more opportunities for peer interaction (especially among girls); and relatively lower levels of household stress due to higher levels of food security in camps, based on universal food support programmes provided by UN agencies and international NGOs. Key informant interviews also suggested that it was easier for programmes to reach camp-based target populations as they were more cohesive and less spread out than households in host communities.

Survey findings across the two rounds of data collection during COVID-19 (for the subset of adolescents successfully surveyed in both rounds) also highlighted that adolescents had more contact with peers and teachers later on in the pandemic (partly due to some easing of lockdowns). Family stress levels and related violence also appeared to have moderated somewhat over time, possibly as the initial shock of the pandemic subsided and communities acclimatized to the containment measures. These signs of recovery notwithstanding, the negative economic shock persisted throughout the pandemic, with only modest improvements overall and continued inequalities among adolescents based on gender, place of residence, household assets, and (to a lesser degree) being out of school. These findings are in line with World Bank and UNHCR macro-level findings, which estimated that Jordan's national gross domestic product (GDP) declined by 8.2% in 2020, where the pandemic's effects have been compounded by economic and political crisis. In terms of poverty rates among Syrian refugees, the same report estimated an 18% increase in households pushed into poverty by the pandemic in Jordan [25].

Our findings have important implications for policy and programming. First, there is an urgent need for investments in adolescent-responsive mental health services to support young people most at risk of depression and anxiety, given that more than 15% of the sample—and especially older girls—are experiencing moderate to severe symptoms of depression and anxiety. Community-based and online counselling services should be prioritized to support these young people, especially in host communities, where the risk of social isolation appears greater [106, 107]. Given that more than a third of adolescents also reported that they lacked access to trusted peers and family members, and could therefore also be vulnerable to mental ill-health in the medium to longer term, investments in low-cost, peer-to-peer support groups that can be maintained through online and (physically distanced) in-person platforms during crisis episodes could be trialled. The finding that many young people reported turning to religion as part of their coping repertoire echoes findings from other Middle Eastern contexts [108]. As such, working with religious institutions could be an entry point to facilitate adolescent-centered support groups. Local volunteer activities for young people could also be expanded—albeit with greater attention to gender dynamics to find ways for girls (who are subject to conservative gender norms around mobility) to participate safely.

Second, the findings suggested a need to support positive coping repertoires and adolescent sources of resilience. As well as accessing opportunities to interact with peers online and offline, findings from the qualitative data underscored that young people valued opportunities to learn new skills and engage in low-cost activities to express their emotions in creative ways (e.g., diary-writing, book discussion groups, drawing). In the same vein, some quick wins might focus on raising awareness about the risks of using tobacco and other substances as a coping mechanism, especially among adolescent boys [109, 110]. It also may be possible to build on lessons from the adolescents who saw a decline in smoking, likely due to financial constraints, and to shape public health messaging about the spill-over benefits (cost savings) of quitting smoking.

Third, adolescents highlighted that household stress (and especially economic constraints) had increased since the pandemic, manifesting in increased intra-household violence,

including violence towards adolescents. These reports suggest an urgent need to address the underlying economic stressors through a scaled package of social assistance [111, 112], and to ensure that linkages to telephone helplines, group-based mental health support and community counselling services are adequately resourced and adolescent-friendly.

Fourth, while the overwhelming majority of young people who were enrolled in formal schooling pre-pandemic were able to continue learning in some way during lockdown, some factors negatively affected engagement with distance education, for boys and girls alike. These factors range from limited internet connectivity and lack of access to devices, to time burdens spent caring for siblings, and other domestic work. Many adolescents also reported limited feedback and support from teachers during remote learning, with boys and refugees especially disadvantaged in this regard. It is therefore important to strengthen the linkages between teachers and students during online learning, and to provide alternative means of learning— e.g., paper-based for those with limited or intermittent internet access and adapted teaching modalities for adolescents with disabilities (e.g. using audio for visually impaired students and sign language interpretation in internet-based instruction)—to minimize educational disadvantage for the most vulnerable adolescents [113, 114]. In terms of returning to in-person learning once schools reopen, younger boys especially were concerned about not being able to resume learning; particular attention should therefore be paid to these cohorts to tackle both education and learning deficits experienced during the pandemic. Equally importantly though —and as studies from other LMIC contexts have emphasized [94]—there is a need to reinvigorate peer interactions and teacher–student communication, as these are vital factors in supporting adolescent psychosocial wellbeing and resilience.

Finally, our findings underscored the importance of increased investment in longitudinal data so as to strengthen the evidence that informs policy and programming to support adolescent health and wellbeing, especially in response to macro shocks such as the COVID-19 pandemic. Investments in additional rounds of data collection would inform an understanding of the longer-term effects of COVID-19 and associated risk-mitigation strategies on the capability trajectories of adolescents in this context.

## Limitations and strengths

Specific limitations and strengths of our analysis are noteworthy. First, while this sample was not nationally representative, it focused on vulnerable adolescents—many of whom are typically excluded from traditional data sets or have limited sample sizes that prevent quantitative analysis in such data sets. Second, while access to the internet is relatively widespread in Jordan, and we provided respondents with phone credit to avoid them incurring out-of-pocket expenses for participation, our sample did have some attrition. Yet rates of attrition were similar to or better than other phone surveys during the pandemic. This was not only due to widespread phone access in our sample, but also likely due to the fact that we followed a strict protocol and called households at least 15 times, across weekdays and weekends, and at different times of day. Attrition rates also were largely non-differential across baseline characteristics, with no differential selection on observables for the sample that combined the COVID-R1 and COVID-R2 data, and limited selection for the panel sample. The use of multivariate regressions and propensity score matching, as well as similar findings across the two samples, assuaged this issue. Third, there may be concerns with misclassification of binary outcomes that may arise from combining data from COVID-R1 and COVID-R2. The fact that findings were qualitatively similar when we restricted to the panel sample somewhat mitigates these concerns. Fourth, phone-based interviews were, by necessity, shorter (especially with younger adolescents), which limited the depth of information that could be elicited. That said, the wide

variety of outcomes measured in our quantitative survey and use of mixed-methods panel data helped overcome this limitation. Fifth, ensuring privacy can be challenging, even in a panel design in which the research team and participants have established trust over time; as such, the adolescent respondents may not have been able to speak candidly (for example, around issues related to illicit drug use or violence given their sensitivity). In the qualitative sample to overcome this we offered flexible scheduling of interviews to better suit windows of time when adolescents had more privacy. Sixth, disentangling the effects of age and calendar time is difficult, given the often fast-paced nature of changes during puberty and adolescence, and the rapidly changing conditions of the pandemic.

## Conclusions

This study presents an in-depth, mixed-methods panel analysis of the dynamic impacts of COVID-19 and associated risk-mitigation strategies on adolescents in humanitarian settings. Our research underscored that the pandemic and related service disruptions have compounded pre-existing social inequalities among adolescents in such settings. Mitigating the impacts of the pandemic on adolescents' developmental trajectories may require multisectoral cooperation and multi-level programmes that attend to priority needs of the most vulnerable adolescents, including food insecurity, gender- and age-based violence, and mental ill-health. Moreover, these programmes should address population-level root causes and determinants of psychosocial wellbeing and resilience of all adolescent girls and boys, given the critical developmental window that the second decade of life offers. Only then can national governments and the international community live up to the promise of the 2030 Agenda for Sustainable Development and its commitment to reducing inequalities (SDG 10) and, ultimately, to leaving no one behind. This commitment must be inclusive of those affected by forced displacement, to ensure that the global community achieves the targets of health (including mental health and sexual and reproductive health) and wellbeing for all (SDG 3.4), a quality education (SDG 4), elimination of all forms of abuse and violence against children (SDG 16.2), and eliminating child, early and forced marriage (SDG 5.3).

## Supporting information

**S1 Table. Determinants of being surveyed at COVID-R1 or/and COVID-R2.**
(DOCX)

**S1 File. Description of GAGE COVID-R1 and COVID-R2 survey outcomes.**
(DOCX)

## Acknowledgments

The authors gratefully acknowledge the survey data collection efforts of the Mindset Jordan team, and the qualitative data collection undertaken by Sarah Alheiwidi, Wafa Al Amaireh, Kifah Bani Odeh, Taghreed Alabbadi and Qasem Shareef. We thank Agnieszka Małachowska for her programme management contributions. We also thank Elizabeth Presler-Marshall, Olivia Simpson and Eric Neumeister for research assistance, and Kathryn O'Neill and Anna Andreoli for editorial support.

## Author Contributions

**Conceptualization:** Nicola Jones, Sarah Baird, Zulfiqar A. Bhutta, Kathryn M. Yount.

**Data curation:** Nicola Jones, Bassam Abu Hamad, Erin Oakley, Jude Sajdi.

**Formal analysis:** Sarah Baird, Erin Oakley.

**Funding acquisition:** Nicola Jones.

**Methodology:** Nicola Jones, Sarah Baird, Manisha Shah.

**Project administration:** Bassam Abu Hamad.

**Writing – original draft:** Nicola Jones, Sarah Baird.

**Writing – review & editing:** Bassam Abu Hamad, Zulfiqar A. Bhutta, Erin Oakley, Manisha Shah, Jude Sajdi, Kathryn M. Yount.

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
