## [Decision Letter · Decision Letter 0]

15 Sep 2021

PONE-D-21-21919Compounding inequalities: Adolescent psychosocial wellbeing and resilience among refugee and host communities in Jordan during the COVID-19 pandemicPLOS ONE

Dear Dr. Jones,

Thank you for submitting your manuscript to PLOS ONE. After careful consideration, we feel that it has merit but does not fully meet PLOS ONE’s publication criteria as it currently stands. Therefore, we invite you to submit a revised version of the manuscript that addresses the points raised during the review process.

ACADEMIC EDITOR:  I request authors to develop a good narrative to link between quantitative analyses with qualitative analyses. Also, this link should be reflected in writing discussion of the paper as well.  Overall, I suggest a minor revision for this paper.

We look forward to receiving your revised manuscript.

Kind regards,

Srinivas Goli, Ph.D.

Academic Editor

PLOS ONE

Journal Requirements:

"The authors wish to gratefully acknowledge the survey data collection efforts of the Mindset Jordan team, and the qualitative data collection undertaken by Sarah Alheiwidi, Wafa Al Amaireh, Kifah Bani Odeh, Taghreed Alabbadi and Qasem Shareef. We wish to thank Agnieszka Małachowska for her programme management contributions. We would also like to thank Elizabeth Presler-Marshall,  Olivia Simpson and Eric Neumeister for research assistance, and Kathryn O’Neill and Anna Andreoli for editorial support. Finally, the authors acknowledge funding for the data collection underpinning the study from UK aid, as well as co-funding from the EMERGE project (Bill and Melinda Gates Foundation Grants: OPP1163682 and INV018007; PI Anita Raj) and Research Grants on Women, Victimization, and COVID-19 (Bill and Melinda Gates Foundation (# INV-003527) awarded through the NBER). "

"Yes

Funding was received from the Research and Evaluation Division of the UK Foreign Commonwealth and Development Office (FCDO) for the Gender and Adolescence: Global Evidence (GAGE) longitudinal study."

Additional Editor Comments (if provided):

I request authors to develop a good narrative to link between quantitative analyses with qualitative analyses. Also, this link should be reflected in writing discussion of the paper as well. Overall, I suggest a minor revision for this paper.

Reviewers' comments:

Reviewer's Responses to Questions

**Comments to the Author**

1. Is the manuscript technically sound, and do the data support the conclusions?

Reviewer #1: Yes

2. Has the statistical analysis been performed appropriately and rigorously? 

Reviewer #1: Yes

3. Have the authors made all data underlying the findings in their manuscript fully available?

Reviewer #1: No

4. Is the manuscript presented in an intelligible fashion and written in standard English?

Reviewer #1: Yes

5. Review Comments to the Author

Reviewer #1: The paper in its current form requires further revisions that are deemed to be minor, and most of these are provided to help improve the paper’s readability and presentation of information. Currently, the paper combines quantitative and qualitative findings spread across eight tables, which seem disjoint in places. While the quantitative methodology is appropriate for the paper, the interlinkages with the qualitative section need to be further developed. Some of the comments below are the reviewer’s reflections that are meant to encourage the authors of this impressive study

6. PLOS authors have the option to publish the peer review history of their article (what does this mean?). If published, this will include your full peer review and any attached files.

Reviewer #1: **Yes: **Dr Danish Ahmad

---

## [Author Response · Author response to Decision Letter 0]

30 Oct 2021

Many thanks for the constructive feedback. Please find below a detailed response to the editor's and reviewers' comments. 

1. Editor Comment

"The authors wish to gratefully acknowledge the survey data collection efforts of the Mindset Jordan team, and the qualitative data collection undertaken by Sarah Alheiwidi, Wafa Al Amaireh, Kifah Bani Odeh, Taghreed Alabbadi and Qasem Shareef. We wish to thank Agnieszka Małachowska for her programme management contributions. We would also like to thank Elizabeth Presler-Marshall, Olivia Simpson and Eric Neumeister for research assistance, and Kathryn O’Neill and Anna Andreoli for editorial support. Finally, the authors acknowledge funding for the data collection underpinning the study from UK aid, as well as co-funding from the EMERGE project (Bill and Melinda Gates Foundation Grants: OPP1163682 and INV018007; PI Anita Raj) and Research Grants on Women, Victimization, and COVID-19 (Bill and Melinda Gates Foundation (# INV-003527) awarded through the NBER). "

"Yes

Funding was received from the Research and Evaluation Division of the UK Foreign Commonwealth and Development Office (FCDO) for the Gender and Adolescence: Global Evidence (GAGE) longitudinal study."

1. Response/Resolution

Thank you for this note; we have made this change.

We have removed this sentence from the acknowledgements and added instead to the Funding Statement. 

Finally, the authors acknowledge funding for the data collection underpinning the study from UK aid, as well as co-funding from the EMERGE project (Bill and Melinda Gates Foundation Grants: OPP1163682 and INV018007; PI Anita Raj) and Research Grants on Women, Victimization, and COVID-19 (Bill and Melinda Gates Foundation (# INV-003527) awarded through the NBER). 

2. Editor Comment

We note that the grant information you provided in the ‘Funding Information’ and ‘Financial Disclosure’ sections do not match. 

2. Response/Resolution

Thank you for this note; we have made this change.

3. Editor Comment

3.Response/Resolution

Thank you for this note; we reviewed the reference list and made revisions/corrections as needed. However, please note that we only added references in order to update the context section and to respond to reviewer comments, rather than retracted references.

4. Editor comment

I request authors to develop a good narrative to link between quantitative analyses with qualitative analyses. Also, this link should be reflected in writing discussion of the paper as well. Overall, I suggest a minor revision for this paper.

4.Response

Thank you for this note—we have revised the narrative to better link the quantitative and qualitative analyses, by providing a clear introductory paragraph to the Results section as follows about the analysis approach: 

To explore the impact of the COVID-19 pandemic on adolescent health and wellbeing and explore the differences in pandemic effects based on pre-existing socioeconomic inequalities, we present our results in four sections. First, we explore pre-COVID-19 vulnerabilities to establish the existing inequalities in our sample prior to the pandemic onset. Then, we present findings on post-COVID-19 disrupted social contexts, describing service closures and livelihood restrictions reported by households in the sample. Next, we explore findings on COVID-19 and adolescent psychosocial wellbeing, focusing on mental health and adolescents’ own concerns about their future during the pandemic. Finally, we examine challenges to coping and resilience under COVID-19, with a focus on both individual-level coping skills and resilience as well as social connectedness and support. Throughout each section, we highlight differences by gender, age, nationality, type of community, baseline assets, baseline school enrolment, and marital status and intertwine quantitative and qualitative findings on each of these topics. 

5. Reviewer 1

Have the authors made all data underlying the findings in their manuscript fully available?

5.Response

The GAGE study is in the process of publicly archiving and making the Jordan baseline and Covid-19 quantitative datasets available to download from the UK Data Archive. We anticipate that these datasets will be published in late 2021.

6. Reviewer 1

The paper in its current form requires further revisions that are deemed to be minor, and most of these are provided to help improve the paper’s readability and presentation of information. Currently, the paper combines quantitative and qualitative findings spread across eight tables, which seem disjoint in places. 

While the quantitative methodology is appropriate for the paper, the interlinkages with the qualitative section need to be further developed. Some of the comments below are the reviewer’s reflections that are meant to encourage the authors of this impressive study

6.Response

Thank you for this note—we have revised the narrative to better link the quantitative and qualitative analyses. We provide more detail next to specific comments on how we create these improved linkages.

7. Reviewer 1

Thank you for the opportunity to review the paper.

The study provides invaluable insights into an important social and global health challenge with the Impact of COVID-19 on vulnerable adolescents living in displaced/temporary residential communities in Jordan. The authors present a mixed-method study using quantitative survey data collected in three rounds with a baseline round in 2018/2019 followed by two rounds of panel data collection in 2020 and 2021, where the effect of Covid induced lockdown on vulnerable adolescent’s health and social development is presented. The survey data was collected as part of an ongoing research program titled ‘Gender and Adolescence: Global Evidence (GAGE)’ which follows close to 3000 adolescents living in camps and host communities in Jordan; while qualitative data were purposively collected from Multivariate regression analysis is conducted on the pooled data comprising young and older adolescents of Syrian, Palestinian and Jordanian heritage on health and social development indicators drawn from the GAGE conceptual framework. The paper supplements quantitative findings with qualitative interviews from the sampled population and have creatively presented both findings. While the paper provides insights mainly from the effect of lockdown on the study population, the underlying pathways reflect the added challenges faced by a vulnerable community that are already disadvantaged due to geo-political and ongoing conflicts in the Middle East. The paper presents opportunities to advance public health interventions and policy targeting communities/adolescents living in vulnerable environments that may interest a wide readership.

The paper in its current form requires further revisions that are deemed to be minor, and most of these are provided to help improve the paper’s readability and presentation of information. Currently, the paper combines quantitative and qualitative findings spread across eight tables, which seem disjoint in places. While the quantitative methodology is appropriate for the paper, the interlinkages with the qualitative section need to be further developed. Some of the comments below are the reviewer’s reflections that are meant to encourage the authors of this impressive study

7.Response

Thank you for your thoughtful comments. We have worked on adjusting the narrative to improve cohesion between qualitative and quantitative findings. 

As discussed above, we added a paragraph at the beginning of the results section to guide the reader as to how we are linking the quantitative and qualitative findings throughout the analysis so as to address this comment.

8. Reviewer 1

Introduction: Introduction from pages 1-3 provides a general context of the impact of covid related lockdowns on adolescent health and wellbeing. The authors seek to imply the effect of educational disruptions and social isolation bought on by the lockdown as the primary pathway affecting wellbeing. While the first three introductory pages set the context, the research population for this paper is introduced much later on page 3 . Even outside of the pandemic, the health and wellbeing of vulnerable adolescents living in settlements camps and under settlements is already expected to be impacted due to disruption of the social determinants of health. The impact of the pandemic hence adds additional burden over and above an existing health risk. The paper currently does not adequately introduce the research population with relevant context. The authors on page 10 make reference to ‘pre-existing social inequalities’; however, the preceding introduction does not substantiate what these are.

8.Response

We have added some additional context on refugee populations and the impact of the COVID-19 pandemic, in addition to the broader information on adolescent health, food security, risk of violence, mobility, enrolment and wellbeing (see edits Introduction, paragraph 1)

Page 3, lines 67-81

9. Reviewer 1

Page 10 (line numbers-not provided in paper) As the pandemic now stretches >2 years; the authors may like to state the data collection dates earlier in relation to the following statement-three and nine months after its onset, on adolescent psychosocial wellbeing and mental health, exploring the role of support systems at the individual, household, community, and policy levels among communities affected by displacement

9.Response

We’ve updated the language in this statement to more accurately reflect the time points of data collection as follows: 

“Using mixed-methods data collected pre-pandemic (baseline conducted in 2018-2019) and at two time points after the pandemic onset (May-July 2020 and November 2020-January 2021), we evaluate the impacts of the pandemic and associated policy responses on adolescent wellbeing and mental health.”

Page 5, line 122

10. Reviewer 1

Methods 

The reference to dates (in pages 10,11) to explain the COVID-19 pandemic trajectory in Jordan was appreciated. However, as infection and case fatality rates are likely to be higher for vulnerable people living in camps where population density is higher, the authors may like to add a few references (if available about death/infection rates) in such communities which are different from the general population

10.Response

We agree—this is interesting and useful information to present for this context. We added some information on the distribution of COVID-19 cases in refugee camps in Jordan. Syrian refugee camps, possibly due to more limited mobility, have consistently lower case rates than the Jordanian population overall. 

Page 6, line 157-165

11. Reviewer 1

Could you please define ‘host communities’ as its relevant to the study and not part of common knowledge?

11.Response

Thank you for this note—we’ve added a footnote to the first in-text reference to “host communities” as follows: 

“Host communities refer to existing Jordanian communities where refugees have settled alongside the local population, outside of formal refugee camp settings. In Jordan, 80.5% of all refugees living in host communities [].”

Page 4, line 114 & footnote

12. Reviewer 1

References are missing on page 5, last para.- Drawing on the work of Amartya Sen [X] and Marta Nussbaum [X]

12.Response

We have updated these. 

13. Reviewer 1

On page 16, table 3, panels a and b provide the paper’s main quantitative findings across different timelines. While the effect sizes are provided, the authors may like to add a note on the underlying measure of association used to obtain the effect size (RR or OR, e.g.). 

13.Response

These measures come from a linear probability model so can be interpreted as a risk difference, or percentage point difference. We clarified what the “effect size” measure means in table three in Panel A and Panel B as follows:

Panel A: For comparisons of baseline wealth, the table presents the effect size, defined here as the coefficient for baseline wealth using a linear regression, and standard error. For binary measures, coefficient can be interpreted as “percentage point difference ,” or risk difference, in the measure of interest for adolescents in households with above-median assets compared to those with below-median assets. 

Panel B: For comparisons of gender, age cohort, school enrolment status, and marital status, the table presents the effect size, defined here as the coefficient for female gender, younger cohort, baseline school enrolment, and being ever-married using a linear regression, and standard error. For binary measures, coefficient can be interpreted as “percentage point difference,” or risk difference. in the measure of interest for adolescents who are female, younger cohort, in-school, or ever married, compared to those who are not. 

14.Reviewer 1

Additionally, some of the measures reported currently don’t have background information provided in the introduction(pages 1-3). This is a repeated comment as I have previously suggested adding few references (if available) in the introduction to provide readers with

some context about the social vulnerabilities encountered by the study’s population

14. Response

We have revised the introduction section to provide additional context for the study population and better introduce social vulnerabilities that are discussed in Table 3 and throughout the paper.

15. Reviewer 1

Results

The transition from quantitative results on page 27 to qualitative results on page 28 is linked well and allows a gradual transition to the qualitative findings under the section on ‘COVID19-and adolescent psychosocial wellbeing’.

15.Response

Thank you for this note. 

16. Reviewer 1

Some of the key characteristics in how ITS, camps and host families differ would provide valuable context to the paper. Adding details in the method or as a section in the

introduction would be important. Page 28, last para reflects on the nature of social networks in ITS as a reason for better PHQ-8 scores. Readers would be better placed to link the insight with a previous description of characteristics as suggested.

16.Response 1

We added some additional detail to the Introduction section to better introduce the reader to the location contexts. We also updated the subsection “Pre-Covid vulnerabilities” to further highlight the baseline differences in the Syrian sample across location (camp, host, ITS) as follows: 

While Syrian adolescents are no more likely to be food insecure or hungry than adolescents of other nationalities, there is substantial variation by location. Syrians who live in host communities and informal tented settlements (ITSs) are more likely to be food insecure than those in camps (58.5% host vs. 56.3% ITS vs. 38.8% camp, p<0.001) and more likely to have been hungry in the past four weeks (19.3% host vs. 22.7% ITS vs. 9.7% camp, p<0.001). Among Syrian households, those living in ITSs are also the least likely to receive food or cash aid (89.7% in ITS vs. 93.1% host vs. 98.6% camp, p<0.001) and to attend Makani adolescent programming (33% in ITS vs. 38.6% host vs. 74.1% camp, p<0.001). Despite these disparities, adolescents in ITS communities are also the least likely to report experiencing household violence in the last 12 months among Syrian refugees by location (38.8% ITS vs. 40.5% camp vs. 51.3% host, p<0.001) and also most likely to report having a trusted friend (76.1% ITS vs. 74.5% camp vs. 69.4% host, p=0.008).

Page 16, line 396-406

17 Reviewer 1

While the paper has collected data from younger and older cohorts in baseline(pre-pandemic), the presentation of results in tables 3 and 4 and in subsequent sections is grouped around residential affiliation (camp versus ITS vs Host family). 

The qualitative insights presented on page 29 provides reflections using age as a reference which is valuable. I found that the quantitative section in table 5(page 31) presents age-based reflections. 

17.Response

We have updated Table 3 to include a comparison by age cohort (see Panel B – Cohort). We do include a covariate for the adolescent being in the younger cohort (age 14 and younger) throughout Tables 4-7. However, we only discuss the differences by this covariate in the narrative where the results are statistically significant. 

18.Reviewer 1

As the results move between qualitative and quantitative sections fairly, I would advise the authors to have a summary section before or soon after the ‘results’ section starts to provide a pathway for readers to follow and improve the section’s linking.

18.Response

We added a brief summary paragraph at the beginning of the “Results” section to orient the reader to the topics covered. 

Page 15, lines 372-383

19. Reviewer 1

The authors have previously mentioned on page 13 that the paper explores whether access to adolescent-specific programs in the form of ‘Makani centres act as a protective factor. This would be a valuable finding to report, especially in R1 and R2 of data collection if the centres were functional. I was unable to adequately tease out the effect of access to Makani centres on the outcomes produced in tables 3-5. I understand that not all residential camps may have the program, but if the authors have looked at it as one of the main interventions available in the camps, it would be valuable to have a section on it in results. This is especially important when reporting qualitative outcomes such as pages 28/39

where adolescents report higher mental health issues. This is relevant to readers understanding the association between availability of access to Makani centres on adolescent mental health (boys/girls and young vs old) during the data collection rounds.

19. Response

Thank you for this comment. We agree that the effect of Makani participation is an important factor—however, as this is an endogenous characteristic for our study population (e.g, people who attend Makani look different than those who do not), we need a more sophisticated methodology to explore its effects robustly. Thus, we explore the impact of Makani participation in Table 8 using a propensity score matching methodology. We also discuss links to adolescent mental health and return to education when schools reopened in the subsection entitled “Policy and programmatic level: social assistance and youth empowerment programming”. 

20. Reviewer 1

The results on page 40 were insightfully presented, especially with regards to the associations of wealth and age on feeling pressured to marry early.

20. Response 

Thank you for the positive feedback. 

21. Reviewer 1

The limited evidence of illicit drug use, as reported on page 43, may also have been due to respondents withholding sensitive information. As qualitative interviews were done virtually, the opportunity to develop rapport between interviewee and interviewer is

limited. However, the role of the social support offered by families may have a mitigating effect on illicit drug use. The influence of family support is evident on page 49 /table 7where family and community coping skills are presented.

21 Response 1

Thank you this is a fair point and we have added a note on this in the limitations. Please do note however as we do in the discussion of limitations that because this is a longitudinal study, the adolescents are already familiar with the research and thus some of the challenges of phone surveys and rapport are lessened as a result. This is especially the case with the qualitative research where the same researchers are doing follow up interviews with the respondents as in previous rounds. 

22. Reviewer 1

The reduced aid offered to families during the pandemic, as explained on page 61 is an important determinant of household wellbeing and vulnerability and may explain

some of the transmitted mental health stresses that are common within a household encountering common stressor(s)

22. Response 

Thank you for this useful note. Yes we agree and have added a sentence reflecting this in the policy and programming section on page 43.

23. Reviewer 1

The discussion and conclusion are well-presented sections

23. Response

Thank you for this kind comment.

---

## [Decision Letter · Decision Letter 1]

8 Nov 2021

PONE-D-21-21919R1Compounding inequalities: Adolescent psychosocial wellbeing and resilience among refugee and host communities in Jordan during the COVID-19 pandemicPLOS ONE

Dear Dr. Jones,

Thank you for submitting your manuscript to PLOS ONE. After careful consideration, we feel that it has merit but does not fully meet PLOS ONE’s publication criteria as it currently stands. Therefore, we invite you to submit a revised version of the manuscript that addresses the points raised during the review process.

ACADEMIC EDITOR: This paper can be accepted if authors makes the minor revisions as suggested by the reviewer. 

We look forward to receiving your revised manuscript.

Kind regards,

Srinivas Goli, Ph.D.

Academic Editor

PLOS ONE

Journal Requirements:

Additional Editor Comments:

This paper can be accepted if authors makes the minor revisions as suggested by the reviewer.

Reviewers' comments:

Reviewer's Responses to Questions

**Comments to the Author**

1. If the authors have adequately addressed your comments raised in a previous round of review and you feel that this manuscript is now acceptable for publication, you may indicate that here to bypass the “Comments to the Author” section, enter your conflict of interest statement in the “Confidential to Editor” section, and submit your "Accept" recommendation.

Reviewer #1: All comments have been addressed

2. Is the manuscript technically sound, and do the data support the conclusions?

Reviewer #1: Yes

3. Has the statistical analysis been performed appropriately and rigorously? 

Reviewer #1: Yes

4. Have the authors made all data underlying the findings in their manuscript fully available?

Reviewer #1: No

5. Is the manuscript presented in an intelligible fashion and written in standard English?

Reviewer #1: Yes

6. Review Comments to the Author

Reviewer #1: I am satisfied with the authors reponses. The authors however are advised to review the references as the clean version shows references without commas and without hypens where multiple references are used, eg(1,2,3,4,5) instead of (1-5)

7. PLOS authors have the option to publish the peer review history of their article (what does this mean?). If published, this will include your full peer review and any attached files.

Reviewer #1: **Yes: **Dr Danish Ahmad(MBBS,MSc,PhD,MNAMS,IPFPH)

---

## [Author Response · Author response to Decision Letter 1]

23 Nov 2021

November 23 2021

Dear Reviewers, 

Many thanks for your additional review of our paper and to Reviewer 1 for confirming that all of their substantive comments were addressed. In response to your request for minor revisions, we would like to note the following: 

i) Regarding the datasets, we have now submitted the Jordan Covid-19 data to the UK Data Archive and it has now been accepted (as of November 22 2021) and will be live within the next month based on standard timelines. 

ii) We have also reviewed the references so that where there are multiple references we have used hyphens rather than listing all numbers sequentially. We have kept this in track changes to show the reviewer the actions taken. 

iii) We have included the requested funding statement: "The funders had no role in study design, data collection and analysis, decision to publish, or preparation of the manuscript."

We hope that we have now addressed all outstanding Reviewer comments 

Thanks and best wishes

Nicola Jones on behalf of the authors

---

## [Editor Report · Decision Letter 2]

10 Dec 2021

Compounding inequalities: Adolescent psychosocial wellbeing and resilience among refugee and host communities in Jordan during the COVID-19 pandemic

PONE-D-21-21919R2

Dear Dr. Jones,

We’re pleased to inform you that your manuscript has been judged scientifically suitable for publication and will be formally accepted for publication once it meets all outstanding technical requirements.

Kind regards,

Srinivas Goli, Ph.D.

Academic Editor

PLOS ONE

Additional Editor Comments (optional):

All the reviewer suggestions have been implemented, thus I am recommending this paper for publication in PLOS One with subject to corrections of reference list inside and outside.
---

## [Editor Report · Acceptance letter]

8 Jan 2022

PONE-D-21-21919R2 

Compounding inequalities: Adolescent psychosocial wellbeing and resilience among refugee and host communities in Jordan during the COVID-19 pandemic 

Dear Dr. Jones:

I'm pleased to inform you that your manuscript has been deemed suitable for publication in PLOS ONE. Congratulations! Your manuscript is now with our production department. 

Kind regards, 

on behalf of

Dr. Srinivas Goli 

Academic Editor

PLOS ONE